# The streaming of plastic in the Mediterranean Sea

Alberto Baudena [1✉], Enrico Ser-Giacomi [2,5], Isabel Jalón-Rojas[3,5], François Galgani [4] &
Maria Luiza Pedrotti[1]

Plastic debris is a ubiquitous pollutant on the sea surface. To date, substantial research
efforts focused on the detection of plastic accumulation zones. Here, a different paradigm is
proposed: looking for *crossroad* regions through which large amounts of plastic debris flow.
This approach is applied to the Mediterranean Sea, massively polluted but lacking in zones of
high plastic concentration. The most extensive dataset of plastic measurements in this region
to date is combined with an advanced numerical plastic-tracking model. Around 20% of
Mediterranean plastic debris released every year passed through about 1% of the basin
surface. The most important crossroads intercepted plastic debris from multiple sources,
which had often traveled long distances. The detection of these spots could foster under-
standing of plastic transport and help mitigation strategies. Moreover, the general applic-
ability and the soundness of the crossroad approach can promote its application to the study
of other pollutants.

[1] Sorbonne Université, CNRS, Laboratoire d'Océanographie de Villefranche, UMR 7093 LOV, Villefranche-sur-Mer, France. [2] Department of Earth,
Atmospheric and Planetary Sciences, Massachusetts Institute of Technology, 54-1514 MIT, Cambridge, MA 02139, USA. [3] CNRS, UMR5805 EPOC,
University of Bordeaux, 33615 Pessac, France. [4] French Research Institute for Exploitation of the Sea (IFREMER), Bastia, France. [5]These authors contributed
equally: Enrico Ser-Giacomi, Isabel Jalón-Rojas. ✉email: alberto.baudena@gmail.com

Since the early 1950s, the global ocean has been contaminated with millions of metric tons of plastic materials[1,2], making plastic a ubiquitous and cumulative pollutant across all the oceans[3,4]. Plastic debris impacts marine ecosystems, entangling or being ingested by marine organisms, acting as vector for invasive species, and absorbing persistent organic pollutants[5–7]. It causes socio-economic damage, harming fishing, navigation, and tourism[1,8,9]. About half the plastic produced globally is lighter than seawater[6,10]. Once in the marine environment, floating plastic debris is transported over large distances by ocean currents[11]. Most eventually ends up in one of the five major subtropical gyres, where it can be retained for decades[12].

The plastic pollution in the Mediterranean Sea is comparable to that in the major plastic accumulation zones[13], making it highly contaminated. Due to its almost closed nature, the Mediterranean Sea retains most of its plastic debris. However, a growing body of evidence indicates that the Mediterranean Sea, in contrast to the major oceans, does not have regions in which plastic debris accumulates[13–15]. It has been suggested that the high spatio-temporal variability in its currents, due to an intense mesoscale activity, riverine inputs of water, wind, and a complex coastal bathymetry, prevents the formation and persistence of such features[14,15]. This gap in our understanding hampers remediation and calls for greater knowledge of plastic dynamics and transport.

In the present work, instead of focusing on areas that are a final destination for plastic debris, we target regions through which large quantities of plastic debris pass (plastic crossroads); plastic debris converges to these crossroads but then passes on. We combine (i) the largest field dataset of plastic concentrations in the Mediterranean Sea to date from the Tara Expedition[16] (called here the in situ plastic concentrations); (ii) the implementation of a recent numerical model, TrackMPD, to simulate the plastic-debris paths at sea[17]; and (iii) a particular Lagrangian diagnostic, the crossroadness, used to identify the plastic crossroads[18]. The Tara Expedition covered, for the first time, the whole Mediterranean basin (122 sampling stations, Fig. 1) with a modern and homogeneous measurement methodology. Readers should refer to Methods for a detailed description of the Tara Expedition, the TrackMPD model, and the crossroadness analysis.

## Results and discussion

**Modeling framework.** Virtual plastic particles were progressively released into the Mediterranean at one-minute intervals between 2013 and 2016. The simulated particles ($N \simeq 1.472 \times 10^8$) are considered representative of floating plastics debris, with the exception of extremely light foamed plastics (such as polystyrene foam) or air filled objects (such as bottles or balls) whose dispersion is mainly driven by windage[19]. These were less than 1% of the debris collected during the Tara Expedition. Virtual particles were released from three types of simulated sources (Fig. 1): (a) 185 coastal cities, with the relative number of particles released at each city proportional to the product of its population[20] and the index of mismanaged plastic waste[2] of its country [50% of the total number of particles; $p_C$]; (b) 200 river mouths, with relative particle numbers proportional to their monthly plastic output [$p_R = 30\%$,[21]]; and (c) discarded at sea, with relative particle numbers proportional to the vessel density at that location [$p_V = 20\%$;[22]].

The trajectories of these virtual particles on the sea surface were simulated until the end of 2017 (Methods). The particles were considered as non-inertial passive tracers, in that they were transported by surface currents (provided to the model at hourly intervals) and by isotropic horizontal diffusion (diffusivity coefficient $K_h$). Surface currents in the simulation included geostrophic and Ekman components, and Stokes drift due to waves, which included indirectly the windage effect[23]. Particles could beach onshore, but only if the local steepness was not greater than 40%[24]. If a particle beached, it could be washed-off, with a probability that decreased exponentially with the time spent on land; this probability depended on a characteristic time scale $T_W$ (Methods). Washing-off events are associated with the presence of storms which, in the Mediterranean, are mainly responsible for land-sea plastic transfer[25]. The simulations therefore mainly depended on the two parameters $K_h$ and $T_W$, and on the proportion of particles released from cities, rivers, and vessels ($p_C$, $p_R$, and $p_V$).

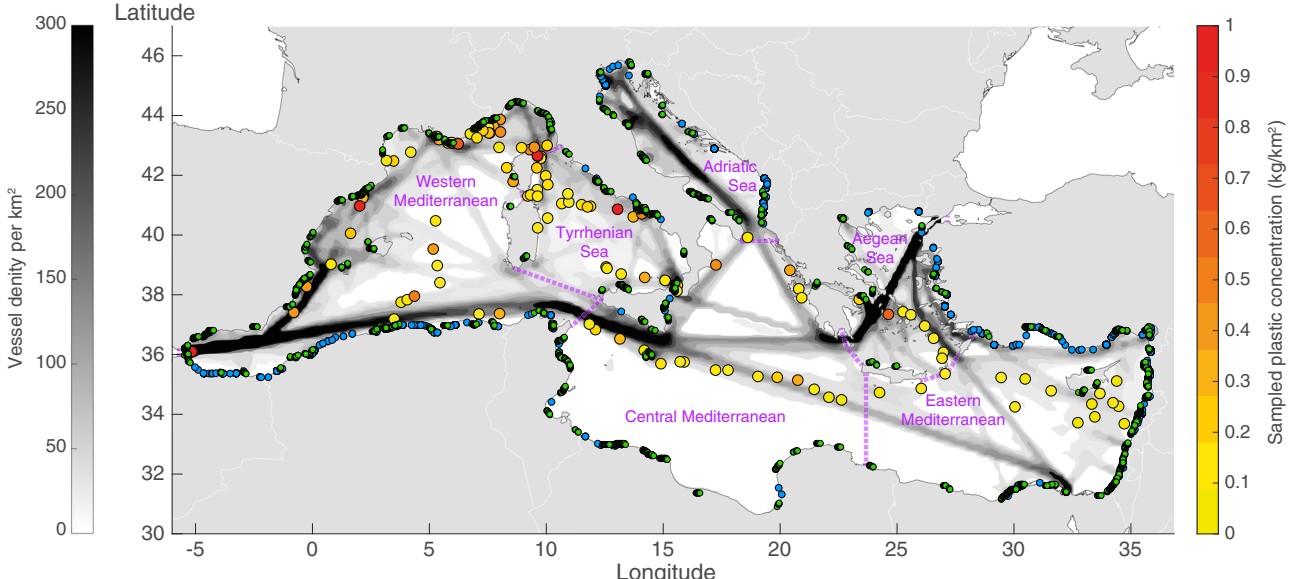

**Fig. 1 Overview of the domain analysed.** Colored circles show the locations of the 122 Tara Expedition stations and the plastic concentrations measured there (right-hand yellow-to-red scale bar)[16]. The green and blue dots near the shore indicate the positions of the coastal cities and river mouths, respectively, used in the model as land sources of plastic. The basin gray scale shows the vessel density (left-hand scale bar), which was used to determine the proportion of plastic debris directly released at sea by vessels. Purple dashed lines separate the different sub-basins.

To assure the robustness of the results, the model was run with 16 different combinations of the two parameters (Methods), consistent with the fact that both parameters are expected to vary in space and time, providing 16 scenarios. Single-scenario results did not differ significantly (Supplementary Note 1, 4, 6, and 8). Therefore, unless specified otherwise, the results reported were obtained from the ensemble average of the 16 scenarios (Scenario M).

**Model validation**. To validate the model, the observed and simulated particle concentrations in the different Mediterranean sub-basins (the western Mediterranean, the Tyrrhenian and Aegean Seas, the central and the eastern Mediterranean; Fig. 1) were compared at the Tara sampling-station locations. The Adriatic Sea was not included in the validation, as no in situ observations were carried out in this basin. Simulated concentrations were calculated from the number of virtual particles around each station on the sampling day (details in Methods). Both observed and simulated concentrations were normalized to permit comparison, so that the sum of all the observed (or simulated) concentrations per basin was equal to 100. The results showed that the model was able to reproduce accurately the distributions of plastic concentration across the different Mediterranean sub-basins (Fig. 2, $p < 0.01$). The highest concentrations were found in the western Mediterranean, the lowest in the eastern Mediterranean, with intermediate values in the other sub-basins. This could be explained by the larger coastal population and anthropic impact in the western basin, but the sampling stations in the western Mediterranean were also generally closer to the coast and to the land-based sources than those in the other sub-basins, including the eastern Mediterranean. The difference between the observed and simulated concentrations in the eastern Mediterranean could be due to the high number of riverine

sources used in the model[21], combined with the large distance of the sampling stations from shore. In this context, the model qualitatively reproduced the distribution of plastic debris as a function of its distance from shore (Supplementary Note 2, Supplementary Fig. 6).

**Plastic crossroadness**. The model was then used to predict the plastic crossroadness in the Mediterranean Sea. For this purpose, a circular neighborhood of radius $\sigma = 11$ km was defined around each point of the domain, disposed on a regular grid of ~15.7 km cell size (Methods). The crossroadness of a given point was defined as the percentage of the total number $N$ of particles that passed through its neighborhood first during the simulation (2013–2017; Methods).

The main plastic crossroads of the Mediterranean were identified from this crossroadness metric as the (small) regions through which large number of virtual particles passed; these were ranked according to the number of particles they intercepted. The first-ranked crossroad was the location with the greatest crossroadness, i.e. the crossroad that intercepted the most particles. The second-ranked crossroad was the location intercepting the most particles once those already intercepted by the first crossroad were excluded. This was applied iteratively until all the virtual particles had been intercepted.

Importantly, it must be noted that 80% of the virtual particles entered the Mediterranean from the 385 land sources (coastal cities and river mouths). By construction, the algorithm will tend to locate the plastic crossroads near these locations. However, this would be a trivial and inaccurate solution, as the model used a limited number of land sources (Methods). For this reason, a buffer zone of 11 km was imposed around each land source, inside which no plastic crossroads could occur (Methods). This is consistent with the fact that the positions and magnitudes of plastic sources are, to date, uncertain. In this way, the crossroads targeted specifically open-sea particles, far from land sources.

The crossroadness distribution (Fig. 3A) was intense close to the Algerian and southern Turkish coasts, and in the Algerian basin, with crossroadness values around 1%. Conversely, low crossroadness was predicted in the Tyrrhenian and Ionian seas. Interestingly, several meanders of moderate crossroadness were present throughout the Mediterranean basin, matching the main circulation features. Two of them occurred on the eastern and western sides of the Adriatic basin, corresponding to the northern and southern currents, respectively.

The first twenty crossroads in the rank order (Fig. 3A) were situated near the coast, generally in proximity to a land source. However, in most cases the intercepted particles were from multiple sources and had often traveled long distances (Supplementary Fig. 1). For instance, 55% of the virtual particles intercepted by the first-ranked crossroad came from Antalya City (Turkey), which was more than 60 km away. An additional 10% originated from the cities of Mersin, Alanya, and Side (4%, 4%, and 2%, respectively), even though they were 420 km, 132 km, and 95 km away, respectively.

Some crossroads mainly intercepted particles discarded from vessels. In this regard, it is remarkable that the sixth-ranked crossroad, located north of Mallorca in the Balearic Archipelago (Fig. 3B, C), was situated far from all land sources (the only nearby city, Palma de Mallorca, was more than 80 km away). Furthermore, the vessel density in its surroundings was not particularly high. Despite this, this particular crossroad intercepted 0.7% of all the particles, mainly vessel discards released in the western Mediterranean basin (Fig. 3B and Supplementary Fig. 1). Surprisingly, 9% of these particles came from Algiers, followed by Barcelona (8.5%) and Valencia (3%), even though

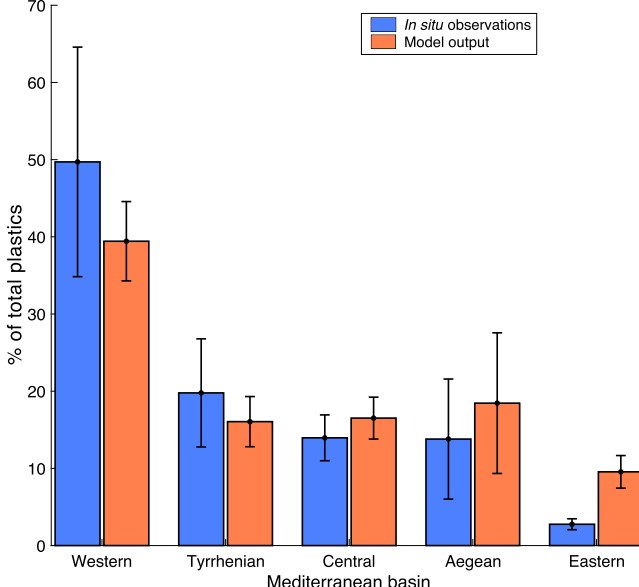

**Fig. 2 Plastic in Mediterranean sub-basins.** Normalized plastic concentrations across the Mediterranean sub-basins from the Tara Expedition in situ measurements (blue columns) and the model predictions (orange columns), with relative uncertainties (standard deviation: black error bars). In situ plastic concentrations were calculated as the debris weight per unit surface area (g/km²). Model concentrations were calculated as the ensemble average (Scenario M) of the number of virtual particles in a prescribed area around each Tara Expedition station (Methods). $R^2 = 0.96$, $p < 0.01$ (Pearson test).

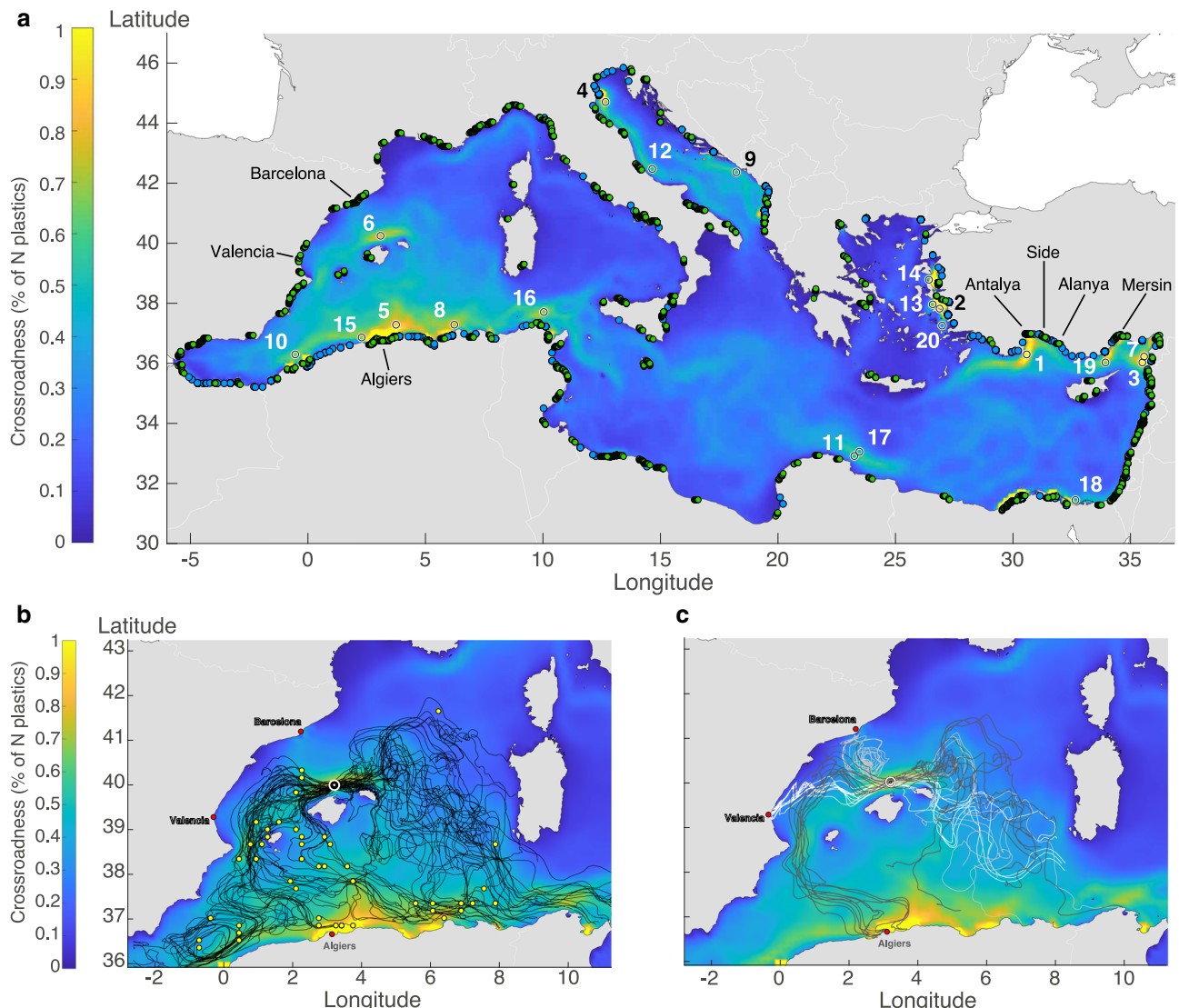

**Fig. 3 Crossroadness and plastic crossroads. a** Crossroadness field calculated from simulated particle trajectories ($N \simeq 1.472 \times 10^8$) between 2013 and 2017. White and black circles, with radius equal to $\sigma$, show the twenty most important plastic crossroads, together with their ranking. **b, c** Focus on the western Mediterranean basin in the region surrounding the sixth-ranked crossroad (white circle), located north of Mallorca. Trajectories of some of the particles passing through this crossroad are shown by the solid lines: (**b**) particles directly released at sea by vessels at the green dots; (**c**) particles originating from three cities: Barcelona (gray lines); Valencia (white); and Algiers (brown).

these cities were 380 km, 150 km, and 250 km away, respectively (Fig. 3C). With the large number of particles intercepted and the rich variety of their origins, this site thoroughly embodied the concept of a plastic crossroad.

The crossroadness allowed us to predict the locations through which high fluxes of plastic debris are expected to pass. The first 20 crossroads intercepted overall ~13% of the virtual particles (Fig. 4) while only covering ~0.3% of the Mediterranean Sea surface. The first 60 crossroads intercepted ~21% of Mediterranean plastics in less than 1% of its surface. To gain a qualitative insight of these percentages, we note that ~1% of the Pacific Ocean surface (the Great Pacific Garbage Patch region[12,26]) contains around 18% of the estimated 117,000 tons of floating plastic in the Pacific[3]. At a global level, the plastics in the Great Pacific Garbage Patch represents ~8% of the estimated total floating plastic debris in around 0.4% of the surface of the world oceans. Even if the crossroadness represents a plastic flux (and not a stock as in the Great Pacific Garbage Patch), and we are comparing relative rather than absolute quantities, these values

gives an idea of the magnitude of particles flowing in the Mediterranean crossroads. Thus, even if persistent plastic accumulation zones are not present in the Mediterranean, we have shown here that a different type of structure seems to exist, crossroad regions through which large amounts of plastic debris transit.

These percentages can be converted into mass fluxes, by multiplying them by the amount of plastic entering the Mediterranean Sea each year (e.g. 100,000 tons[15]) and the number of years particles were released (4). Thus, a crossroadness value of 1% would mean 4000 tons transiting during the simulation period.

The locations of the crossroads seem to be intimately connected with the anthropogenic pollution sites and with the circulation features transporting the debris. In this regard, the fact that most crossroads lay in coastal areas suggests that boundary currents play a pivotal role in determining their location. These circulation structures can collect large quantities of plastic debris released from land, funnelling and carrying them over large distances.

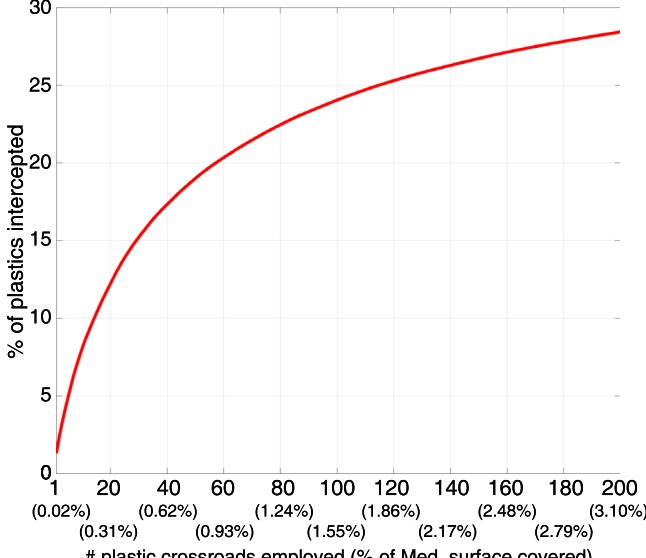

**Fig. 4 Particles intercepted by the plastic crossroads.** Cumulative percentage of the $1.472 \times 10^8$ virtual particles intercepted as a function of the number of crossroads considered. The quantities in parentheses are the percentages of the Mediterranean Sea surface covered by the crossroads collectively.

By construction, all the particles crossing the land-source buffer zones were intercepted by the crossroads. These constituted 33% of all the particles released; the remaining 67% did not leave the buffer region, and so could not reach the crossroads. This suggests that particles released in coastal areas are strongly retained nearby, in agreement with previous studies[15,27].

**Plastic beaching and surface sinking rate**. To quantify this retention in the coastal zones, which can mean beaching there, we calculated from the model the net beaching rate along the coastline: the net number of particles deposited daily per kilometer of beach (the difference between the numbers of beached and washed-off particles; Methods). We considered the number of particles released $N$ as representative of 100,000 metric tons of plastic released per year. This value has been adopted in a previous study focused in the Mediterranean[15], and represents a compromise between recent estimates, both lower[27] and larger[28,29]. However, this choice does not affect the pattern obtained (Methods). The largest net beaching rates (Fig. 5) were found on the Egyptian coast and in the central part of the Algerian coast (43–47 kg/km/day). High values were also predicted in the Cilician basin, along the Syrian coast, and in the Po Delta region (10–14 kg/km/day). The coasts least affected by plastics were in the southern part of Crete, in the Gulfs of Taranto (Italy) and Lion (France), and on the eastern Sardinian sector (all about 0.3 kg/km/day). The net beaching rates predicted on Corfu [(1.9 ± 2.3) kg/km/day] were very similar to the observed values [(1.9 ± 2.2) kg/km/day[30]], even if the predicted values would change when changing the amount of plastic released per year. Comparing the values obtained in other regions with observations was not possible, as most of the time only occasional measurements are carried out on Mediterranean beaches. For those cases, comparison was difficult because the amount of plastic debris on a given beach at a given time depends on past events such as previous beach clean-ups, beachgoer activities, and current seasonality. However, we found good qualitative agreement with such measurements across the basin (Supplementary Note 9).

The model was run again neglecting the slope of the shoreline (Scenario NSS) to determine the importance of this parameter; there was a reduced agreement with the in situ data (Supplementary Note 7). This highlights the relevance of shoreline slope in modeling beaching and, ultimately, for plastic tracking.

Deposition on the shore is not the only fate of plastic debris; it can leave the surface due to sinking. One of the main processes inducing sinking of plastic is the biofouling effect[19]; colonization of plastic debris by marine organisms such as plankton or algae increases the relative density of a particle, causing it to sink. The period of time necessary to induce sinking is called the biofouling time. Here, we analysed the trajectories of the 16 scenarios to determine the number of particles in a square kilometer of surface sinking each day (the surface sinking rate). We assigned the same mass to each particle. The fraction of the mass considered as fully biofouled increased progressively according to a logistic biofouling probability (Methods), used in previous studies to describe algal colonization[27,31]; this peaked at the biofouling time. As the time spent in the water approached the biofouling time, the fraction of the mass which sank increased exponentially (Supplementary Fig. 2). However, a value for the biofouling time is still a matter for discussion[31,32]. Evidence suggests that different biofouling times are expected, depending on particle size, pristine density, and biotic factors[19,33,34]. For these reasons, we used four biofouling times (from 50 to 200 days, Supplementary Fig. 2), and averaged the results.

The largest surface sinking rates (>40 g/km²/day) were predicted in the Cilician basin, especially in Mersin Bay and in the Gulf of Antalya, where several plastic sources are present (Fig. 5). Other regions affected by considerable surface sinking rates were the Adriatic basin and the western Mediterranean, in particular the Balearic Archipelago sector. The Gulf of Lion, the Gulf of Taranto, and the Aegean Sea had the lowest surface sinking rates.

Comparing this metric quantitatively with observations was not possible, due to the lack of suitable measurements. The only (qualitative) comparison possible is with seafloor plastic concentrations. The complexity of this task is exacerbated by the fact that where debris leaves the surface can be significantly different from where it reaches the bottom[35]. Despite the complexity of these dynamics, our predictions agreed qualitatively with bottom surveys of plastic debris (Supplementary Note 10).

Beaching and biofouling significantly impacted the budget of floating plastic debris, with the former playing a paramount role: around 87% of the particles were beached by the end of the release period, while around 12% were fully biofouled (Supplementary Note 2, Supplementary Fig. 7). The net beaching and surface sinking rates found here are similar to previous estimates[15]. Local differences are explained by the diverse nature of the sources, and by the beaching and washing-off dynamics in our model (Methods and Supplementary Note 3).

**Sensitivity analyses and potential applications**. The results presented here did not change significantly across the 16 scenarios simulated (Supplementary Note 1.1, 4, 6.1, 8.1), neither when changing the proportion of particles released from cities, rivers, and vessels ($p_C$, $p_R$, and $p_V$; Supplementary Note 1.2, 6.4, 8.2). Neither did the buffer distance from land sources nor the neighborhood radius affect the crossroadness analyses significantly (Supplementary Note 6.2, 6.3). Even though it was not taken into account directly, the crossroadness results were not affected by the biofouling process either because most of the virtual particles were intercepted in their first days at sea, when colonization was still limited (Supplementary Note 6.5).

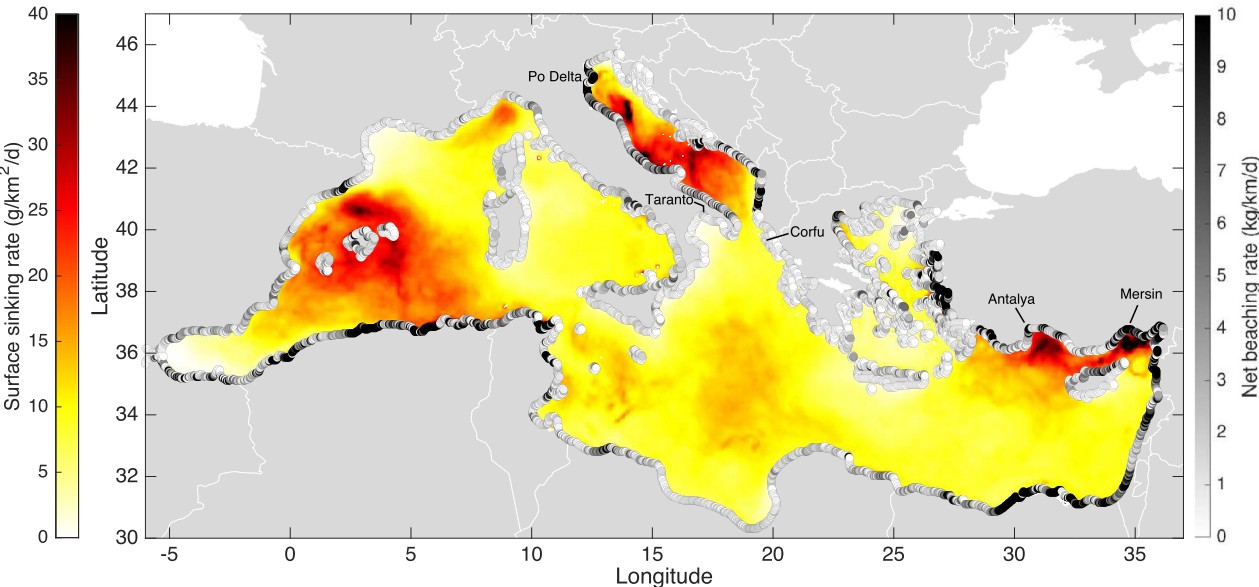

**Fig. 5 Mean net beaching and surface sinking rates in the Mediterranean Sea between 2014 and 2016.** Coastal gray dots show the net amount of plastic debris (kg) beached daily per kilometer of shoreline (gray scale at right). The surface sinking rates are the amounts of plastic debris (g) sinking in a square kilometer of surface each day due to biofouling (color scale at left).

The location of some crossroads may change if the plastic sources change but their existence would most likely continue, together with the number of particles they intercept. This is because of (i) the robustness of our analyses, in particular with respect to the proportion of particles released from cities, rivers, and vessels, and the size of the buffer zones around the land sources; indeed, the use of such buffer zones implies indirectly the need to associate an uncertainty with the land sources. In addition, (ii) crossroadness and crossroads are intimately connected to circulation patterns, and have been proved to be reliable in intercepting real drifters[18]. The adaptability and accuracy of the method here could lead to its application to other regions and time windows, or into the study of other pollutants or passive tracers[36]. At a global level, comparison of the plastic crossroads with the major world plastic-accumulation zones could add to knowledge on plastic transport, spreading, and accumulation. At regional scales, this method could help design a network of fixed stations to monitor and eventually remove plastic items from the sea surface (e.g. https://theoceancleanup.com/), especially in basins where plastic debris does not accumulate[37–40].

The fragmentation of virtual particles into smaller debris was not included in our model, as no quantitative description of this process is available to date. Future studies need to address this question, as 80% of microplastics found at sea do not enter directly the marine environment but are from the disintegration of larger items. Further research is essential to better evaluate the temporal variability of the relative contributions of cities, rivers, and vessels[41]. For instance, coastal populations increase during summer due to tourism[42], with supposedly larger amounts of plastic discharged into the sea. In addition, further data are crucial to improving the calibration of the model (for instance, with a bayesian approach[27]) and the description of beaching and washing-off dynamics. These could benefit from higher-resolution datasets reproducing the coastal dynamics more accurately, together with the hydrodynamic field responsible for plastic transport. To achieve rigorous validation of surface sinking and net beaching rates, future empirical studies should provide time series of washed-up and bottom-deposited plastic volumes. The description of surface sinking rates could benefit from

considering spatial variation in biofouling rates[34,43]. Furthermore, three-dimensional modeling is required to investigate the fate of plastics from surface to seafloor.

Finally, the ecological implications of the presence of crossroads should be investigated. Although they are not necessarily zones of high plastic concentration, they are regions in which wildlife exposure to plastic at a given time might be relatively low, but cumulatively high. For example, in fish nursery areas, the presence of toxin-contaminated plastics can impact the survival of fish larvae during this very vulnerable stage of their life cycle, with significant socio-economic consequences[44]. In addition, the circulation structures responsible for their existence could drive the dispersion of plastic pollutants into remote or marine-protected areas, impacting aquatic biota critically.

## Methods

**In situ plastic measurements from the Tara Expedition.** Microplastic samples were collected in Mediterranean Sea waters during the Tara Expedition[16], which was conducted between May and November 2014 (Fig. 1). Plastic items were collected at 122 stations in manta nets (height 25 cm, width 60 cm, mesh size 333 μm). These were towed at an average speed of 2.5–3 knots for 60 min over a mean distance of ~4 km. The items collected at each station were counted, and their size, weight, and surface area measured. By combining this information with the water volume sampled through the manta net, different types of plastic concentrations were calculated at each station (further details in[16]). Eight categories were defined: (1) particles of size up to 5 mm, g/km²; (2) particles of size 5–20 mm, g/km²; (3) particles of size larger than 20 mm, g/km²; (4) all particle sizes, g/km²; (5) particles of size up to 5 mm, items/km²; (6) all particle sizes, items/km²; (7) total surface area of plastic debris per unit of sea surface, m²/km²; (8) surface area covered by plastic fragments per unit of sea surface, m²/km². The plastic concentrations were regrouped according to the sub-basin in which the stations were located (Fig. 1). Five sub-basins were considered: the western, central, and eastern Mediterranean, and the Tyrrhenian and Aegean Seas. The Adriatic sub-basin was excluded, as no sampling was carried out there. For each of the 5 sub-basins, mean values were obtained for each of the 8 categories, together with the standard error.

**Settings in the Track-MPD model: plastic release and sources.** Virtual plastic particles were released into the Mediterranean at one-minute intervals between January 1, 2013, and December 31, 2016. Three type of plastic sources were considered, in line with previous works[12,15]: main coastal cities; river mouths; and directly released at sea from vessels. 50% of the particles were released from cities, 30% from rivers, and 20% from vessels, based on the proportions used in previous Lagrangian models tracking plastic debris[15]. The descriptions of the three different types of sources are as follows.

- City positions and populations (2015 data), were downloaded from the Urban Cities Database of the Global Human Settlement database[20]. Cities with more than 50 thousand inhabitants and less than 20 km from the coast were selected (185 in total). In each city, a number of coastal locations were selected from which the virtual particles were released. This selection took into account the projection of the city shape (provided in the database) along the boundaries of the hydrodynamical field. The number of the particles released at each city was proportional to the product of its population and the index of mismanaged plastic waste per inhabitant of the corresponding country (following[2]).

- Rivers: we used the estimates of plastic release for each Mediterranean river in[21]. This dataset provides the monthly variations in the amount of plastic entering the sea from rivers, estimated from river watersheds and runoffs combined with population density and the index of mismanaged plastic waste, and averaged for the period 2005–2014. The 200 most polluting rivers were selected from this dataset. Finally, virtual particles were released from each river from a cloud of points at its mouth (and not from a single point), randomly distributed over a surface of area proportional to the mean runoff.

- Vessels: the probability of virtual particle release at a given point was set proportional to the vessel density at that point, estimated by[22] for 2015 at a spatial resolution of 1/6°. This resulted in a total of 9033 points in the Mediterranean Sea as sources of particles released by vessels in the Mediterranean Sea. Note that the vessel density includes fishing-vessel trajectories but not the fishing activity, which in some cases is responsible for the totality of plastic debris found on the seafloor[45]. However, this is not the case in the Mediterranean, where fishing activity, constrained by the limited continental shelves (Gulf of Lion, northern Adriatic Sea, Maltese waters, etc.), mainly releases plastic items directly on the seafloor[46,47].

**Velocity field and trajectory computation.** The velocity field used to simulate the transport of particles between 2013 and 2017 was obtained through the combination of two hydrodynamical fields, both downloaded from the Copernicus Marine Environment Monitoring Service (CMEMS, http://marine.copernicus.eu/). The first product was the MEDSEA_REANALYSIS_PHYS_006_004 (Med_RE), which provides horizontal currents at the surface, and includes geostrophic and Ekman components. It has a spatial resolution of 1/16° and a temporal resolution of 1 day. The second product was the MEDSEA_HINDCAST_WAV_006_012 (Med_HI); it provides the Stokes drift (not provided in Med_RE). It has a spatial resolution of 1/24° and a temporal resolution of 1 h. The two velocity fields were combined as follows. The Med_RE data, with a temporal resolution of 1 day, were interpolated at an hourly frequency to align with the temporal resolution of the Med_HI data. The hourly interpolations of the Med_RE data were then spatially interpolated over the grid of the Med_HI. These two fields were then summed together, giving the final velocity field. This had a spatial resolution of 1/24° and a temporal resolution of 1 hour. This combination allowed us to take into account the geostrophic component, the Ekman effect, and the Stokes drift.

The TrackMPD model reads the velocity field offline, and uses it to compute particle trajectories. These are calculated with a Runge-Kutta scheme of order 4–5 in both time and space, with a time step of 6 minutes (refer to[17] for further details). In order to approximate the small-scale turbulence dynamics not represented in the hydrodynamical field, horizontal diffusion was taken into account. Its intensity was represented by the coefficient of horizontal diffusion $K_h$. Kaandorp et al.[27] tested $K_h$ values ranging from 1 to 100 m²/s, and obtained an optimal $K_h$ estimate of 10 m²/s, in line with those used in previous plastic studies[15,17]. For this reason, we used four $K_h$ values close to this estimate: 0, 5, 10, and 15 m²/s.

**Beaching description.** Haar et al.[24] observed that no significant plastic accumulation was found on shores with steepness greater than 35%. This result was included in the beaching condition of our numerical model by considering the shoreline gradient. When a particle reached the shoreline, it had two possibilities: if the steepness of the closest shore was less than 40%, it was deposited on the shore and considered as beached; otherwise, it returned to its last position in the water.

A shore-gradient map of the European coast of the Mediterranean with a spatial resolution of 25 m was obtained from Copernicus Land Monitoring Service (European Environment Agency, EEA). For the Asiatic and African coasts, a shore-gradient map was obtained from a topographic map provided by the US Geological Survey website (https://earthexplorer.usgs.gov/). This also had a spatial resolution of 25 m. In total, the shoreline gradient was extracted for 224,194 shore points. 15% of the Mediterranean coasts have a shore too steep to permit beaching.

The deposition of debris on Mediterranean beaches is closely connected to storm events[25]. Storms were implicitly taken into account by the velocity field, which included the Stokes drift (which depends on wave height and direction).

**Washing-off description.** Particles deposited on the beach have the possibility of being washed off and resuspended. We associated washing-off events with the presence of storms, which play a pivotal role in the Mediterranean beach-litter turnover rate[25]. Tidal effects on particle resuspension were not considered in the

present study, as tidal ranges in the Mediterranean are very small (generally less than 0.5 m in amplitude).

Firstly, a time series of significant wave heights was produced for 6169 points along the Mediterranean coast. The time series encompassed the period of advection (between January 1, 2013 and December 31, 2017), with a temporal resolution of 1 hour. The significant wave heights were obtained from Med_HI. We assumed that a shore was subject to a storm event when the significant wave height at that shore was larger than a prescribed threshold. This threshold was taken to be the 95 percentile[48] of all the significant wave heights along all the Mediterranean coasts in 2007 (1.69 m). The number of storms per month obtained in this way (Supplementary Fig. 3) was in line with the beaching and resuspension events observed on the Israeli coast[25]. Storm frequency has a seasonal cycle: an increased number of events occurs during winter and early spring, fewer during summer.

With a storm present at the location of a given beached particle, the probability $P$ of the particle washing-off depended on the time $t_B$ already spent on the beach:

$$P = 0.5\, e^{-\frac{t_B}{T_W}}. \tag{1}$$

$T_W$ is the half the mean time spent by a particle on the beach (half-life) before being resuspended again. $P$ decreases exponentially with $t_B$, as assumed in previous plastic-modeling studies[15,17]. After being beached for a time longer than $2T_W$, the chances of a particle being washed off became negligible. We chose $T_W$ values spanning 50–200 days. In a previous work, this value was set to two days[15], based on oil-spill studies. However, an oil spill behaves differently from plastic debris, especially once it reaches the shore. Here, we justified the $T_W$ values with the findings of Bowman et al.[25], which evaluated the in and out fluxes of plastic debris on five Mediterranean beaches. By assessing the litter turnover rate, they found that debris can be resuspended even several months after its arrival. Importantly, two out of the five beaches surveyed in that study had no human access, thus excluding any bias due to beach cleanups or beachgoer pollution.

**Biofouling description and calculation of the surface sinking rate.** The process of colonization of floating plastic debris by marine organisms such as plankton or algae is usually referred to as *biofouling*. This increases the relative density of the plastic debris, and can induce its sinking even if the plastic density is less than that of seawater. The period of time necessary for colonization of a plastic particle to cause sinking is the *biofouling time*. The probability $P_s$ of a particle sinking as a function of the time $t$ passed since it first entered the water was given by a logistic function, used in previous studies to characterize biofouling[27,31]:

$$P_s(t_W(t)) = \frac{1}{1 + e^{r * (t_W(t) - T_{BF})}} \tag{2}$$

$t_W(t)$ is the total time spent in the water (which can be less than $t$ due to beaching events), $T_{BF}$ is the biofouling time, and $r$ is the slope of the probability curve at its inflection point. Each particle represents a certain mass of plastics $m$. At a given time, the mass of biofouled plastic $m_B$ is given by the product of $P_s(t_W)$ and the quantity of plastic yet not biofouled. Thus, based on Eq. (2), we could estimate the amount of plastic biofouled at each time step of each trajectory (Supplementary Fig. 2). The surface sinking rate was expressed as g/km²/day.

Biofouling was not calculated directly during the runs of the TrackMPD model, but was calculated offline from the output trajectories. This was for two reasons. The first is that there are very few studies providing information on the biofouling time. Fazey et al.[31] observed a biofouling time of 12 weeks. However, ~50% of plastic particles studied by Kaiser et al.[32] did not sink even after 14 weeks. The second reason comes from theoretical predictions of the biofouling behavior of plastic particles. Kooi et al.[33] and Chubarenko et al.[19] showed that the biofouling is affected by several physical constraints (such as particle size, shape, and density) and factors affecting the colonization process (such as light, temperature, and algal growing capacity). For instance, under the same conditions, a sphere of 2 mm has a biofouling time six times greater than that of a sphere of 1 mm radius. Given these considerations, a large range of biofouling times is expected to occur. The offline computation allowed us to test different biofouling times. Finally, we note that the identification of the plastic crossroads, which represented the main objective of the present work, was not consistently affected by the biofouling process (Supplementary Note 6.5).

Using the amount of floating plastic entering the Mediterranean annually as 100,000 tons[15], we calculated the surface sinking rate over the whole Mediterranean. Note that the surface sinking rate pattern does not depend on the amount of plastic entering the Mediterranean, but only its intensity does. For instance, assuming a 30-fold decrease[27] or 2-fold increase[28] would change the surface sinking rate proportionally, but not its pattern. The surface sinking rate calculation was done using four different biofouling times $T_{BF}$ (50, 100, 150, and 200 days). We set $r = 1/3$, so that the shape of the probability function curve was consistent for the four $T_{BF}$ values (Supplementary Fig. 2). With this $r$ value, the fraction of biofouled mass increased from 0.1 to 0.9 in a period of time of ~7 days centered on $T_{BF}$. The four surface sinking rates, calculated with the corresponding $T_{BF}$ values, were averaged together, providing the final field.

**Calculation of the net beaching rate.** The beaching rate was calculated by multiplying the number of virtual particles deposited on a given section of shore during a given time period by the particle mass (obtained from assuming that 100,000

metric tons of plastic enters the Mediterranean each year[15]), then dividing by the shore and time-period lengths. Analogously, the washing-off rate was derived from the number of particles resuspended from a given shore length over a prescribed period. Finally, the net beaching rate was calculated as the difference between the beaching and washing-off rates. All these metrics were expressed as kg/km/day. As for the surface sinking rate, the net beaching rate pattern does not depend on the amount of plastic entering the Mediterranean. We note that the TrackMPD model was run without including biofouling explicitly. Therefore, some of the particles considered as beached in practice never reached the coast, as they sank due to biofouling. To identify them, we considered the total time spent in the water by the virtual particles after their release into the Mediterranean. If, at a given time $t_i$, the time spent in the water exceeded the biofouling time $T_{BF}$, the particle was considered as fully biofouled, and was excluded from the counting from that time on. The $T_{BF}$ values were the same as those used for the calculation of the surface sinking rate (50, 100, 150, and 200 days). Thus, four net beaching rates were calculated, one for each biofouling time, and were averaged together.

The surface sinking and the net beaching rates refer to the 2014–2016 period. 2013 and 2017 were excluded, as the model results were incomplete: the former missed particles released in 2012, the latter did not considered particles released in 2017.

**Parameters chosen and number of particles released**. The model runs depended on two parameters: $K_h$ and $T_W$. To test the sensitivity of the results with respect to these parameters, four $K_h$ (0, 5, 10, and 15 m²/s) and four $T_W$ values (25, 50, 75, and 100 days) were combined, providing 16 model scenarios. If not specified otherwise, the results reported refer to the ensemble average of the outputs of the 16 scenarios (Scenario M). The sensitivity of the results with respect to the proportion of particles released from cities, rivers, and vessels ($p_C$, $p_R$, and $p_V$) and to the biofouling process were simulated offline. The results of the robustness analyses are provided in Supplementary Note 1, 4, 6, and 8. The sensitivity of the crossroadness with respect to the biofouling process is reported in Supplementary Note 6.5.

A supplementary scenario was modeled in which the shore steepness was not taken into account (Scenario NSS), with $K_h = 0$ m²/s and $T_W = 50$ days. This was to evaluate the effect of rocky shores on the results. Its outputs, which are reported in Supplementary Note 7, were not used for the Scenario M computation.

For every simulation, around $9.2 \times 10^6$ particles were progressively released between January 1, 2013 and December 31, 2016. In total, $\sim 1.564 \times 10^8$ particles were released, of which $N \simeq 1.472 \times 10^8$ were used for Scenario M. Particles were advected for one year plus the time necessary to reach the end of a month. For instance, two particles, released respectively on January 1 and January 15, were both advected until January 31 of the following year. As a consequence, the final day of simulation was December 31, 2017. The advective time period was on average $\sim$380 days. The effect of the advective time on the accuracy of the simulations is given in Supplementary Note 5. Crossroadness analyses were not affected by the length of the advective period, as a large majority of the trajectories were intercepted in their first few days in the water (Supplementary Note 6.5 and Supplementary Fig. 9).

**Estimation of virtual particle concentrations corresponding to the in situ data**. For each in situ station, we defined an area representing the water sampled at that location. The area was a stadium shape, with two semi-circles of radius $r_S$ centered respectively on the starting and final points of the manta transect. The number of virtual particles in each of the 122 stadium shapes on the day of sampling was counted for every scenario. For Scenario M, the number of virtual particles at a given station was the sum of the quantities obtained across the 16 scenarios at that location. Different stadium radii $r_S$ were tested, ranging between 0.02° and 0.2°. The best correlations between predicted and in situ concentrations were obtained with $r_S = 0.05°$ ($\sim$5 km), which was the value used here. This value is consistent with the mean length of the manta transects ($\sim$4 km). Finally, the 122 virtual concentrations were regrouped according to the sub-basin partition reported in Supplementary Note 1.1, and the mean values and standard errors calculated for each sub-basin.

**Crossroadness definition and identification of the main plastic crossroads**. Here, we adopted the approach proposed by Baudena et al.[18], with some modifications. The crossroadness was calculated from the trajectories of the virtual particles in the following way. First, a regular grid of points (spaced 15.7 km both latitudinally and longitudinally) was created over the whole Mediterranean domain (10,255 points in total). A circle of radius $\sigma \simeq 11$ km (0.1°) surrounded each grid point. Then, the number of trajectories $N_i$ passing at least once through the circle around the $i$th grid point during the simulation period (2013–2017) was counted. Finally, the crossroadness $CR_i$ of the $i$th grid point is given by:

$$CR_i = \frac{N_i}{N} \cdot 100 \tag{3}$$

where $N$ is the total number of particles released ($\sim 9.2 \times 10^6$ for each of the 16 scenarios). For Scenario M, $N \simeq 1.472 \times 10^8$ and $N_i = \sum_{j=1}^{j=16} N_{ij}$, where $N_{ij}$ is the number of trajectories simulated in the $j$th scenario passing inside the $i$th circle.

The crossroadness of the $i$th domain point is the percentage of all the $N$ virtual particles which passed through its neighborhood first.

In identifying the main plastic crossroads, a buffer of $\sim$11 km (0.1°) was imposed around each land source. Grid points whose circle partially or totally overlapped a buffer were excluded. This led to a total of 9193 grid points as potential crossroad locations. The first-ranked plastic crossroad was the grid point with the highest crossroadness value. The second-ranked crossroad was that intercepting the most virtual particles, once those intercepted by the first crossroad were excluded. Analogously, the third-ranked crossroad was determined once all the particles intercepted by the first two were excluded. This was iterated until all the particles were intercepted, leading to a series of plastic crossroads.

## Data availability

The velocity field and wave height products used to run TrackMPD model are available on the E.U. Copernicus Marine Environment Service Information website (CMEMS, http://marine.copernicus.eu/). The Mediterranean shore steepness is available on Copernicus Land Monitoring Service (European Environment Agency, EEA) and on the US Geological Survey website (https://earthexplorer.usgs.gov/). The crossroadness and plastic crossroads maps, the detailed origin of the plastic debris intercepted by the plastic crossroads, the plastic sources distribution, as well all the data necessary to produce the figures of the Main Text and Supplementary Note are available at https://doi.org/10.5281/zenodo.5931213. The in situ plastic concentrations are available at https://doi.org/10.5281/zenodo.5538237.

## Code availability

The TrackMPD code is available at https://github.com/IJalonRojas/TrackMPD.

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

## Acknowledgements

We thank the commitment of the following institutions: CNRS, Sorbonne University, LOV, Genoscope/CEA. The Tara Ocean Foundation and its founders and sponsors: agnès b.®, Etienne Bourgois, the Veolia Environment Foundation, Lorient Agglomeration, Serge Ferrari, the Foundation Prince Albert II of Monaco, IDEC, the "Tara" schooner, crews and teams. We thank MERCATOR-CORIOLIS and ACRI-ST for providing daily satellite data during the expedition. We are also grateful to the French Ministry of Foreign Affairs for supporting the expedition and to the countries that graciously granted sampling permission. The authors are grateful to John Dolan, Georgios Fylakis, Gaby Gorski, Gerasimos Korres, Marie-Emmanuelle Kerros, Marthe Larsen Haar, Svitlana Liubartseva, Paola Proietti, Sara Sergi, and Romain Troublé for their helpful advice and information provided on the dataset. We also thank Nicolas Benoit for continuous assistance on the Mesu computational server, and Peter McIntyre for the english revision of the manuscript. This study is part of the "PlastiMed BeMed : Closing the plastic tap" project conducted by the International Union for Conservation of Nature (IUCN) with the financial support of the Prince Albert II of Monaco Foundation. E.S-G. is very grateful for support from the Simons Foundation: the Simons Collaboration on Computational BIOgeochemical modeling of Marine EcosystemS (CBIOMES #549931).

## Author contributions

A.B. designed the research with assistance from E.S.-G., I.J.-R., and M.L.P. I.J.-R. developed the first version of the TrackMPD model, which was optimised by A.B. A.B. run the TrackMPD model, analysed the data, and wrote the paper. M.L.P. and F.G. provided the data. All authors provided feedback and helped in shaping the manuscript.

## Competing interests

The authors declare no competing interests.
