## [Peer Review File · Nature Communications]

Title: The streaming of plastic in the Mediterranean seaREVIEWER COMMENTS

Reviewer #1 (Remarks to the Author):

Review of "The streaming of plastic at sea" by Baudena et al

In this article, the authors combine an extensive observational dataset, state-of-the-art simulations and a neat novel Lagrangian analysis tool to investigate the dynamics of floating plastic in the Mediterranean. This is a very relevant, well-written and original manuscript, that further pushes the understanding of an important question: the plastic floating in the Mediterranean Sea. I therefore endorse this manuscript, but encourage the authors to make a few changes for better readability

Major comments:

- 1) I do not think the comparison to the plastic concentration in the North Pacific makes much sense. The authors are comparing a stock to a flux, which is a bit like apples and oranges. If they really want to make this comparison then I suggest they also calculate crossroads in the Pacific; otherwise I would strongly advice to leave out of the manuscript here. There is plenty of other good material anyways
- 2) The relative weighing of the different sources seems a bit arbitrary and ad hoc. It doesn't matter too much for the crossroads analysis itself, I presume. But the weighing would matter for the analysis of total amount of beached and biofouled plastic. It is also somewhat confusing that the crossroads analysis is unites (i.e. only based on particle counts), whereas the later analysis is in mass units. Why is that? Why not choose either particles or mass throughout the manuscript? And why do the authors pose that 100,000 tons of plastic enters the ocean, when other papers give other estimates? Why not estimate the total flux by comparing their own model and observations?
- 3) Do the authors have any idea why their model agrees (to within error bars) with the observations in all basins except the Eastern basin? What might be going wrong in the Eastern Basin?

Minor comments:

- line 1: The title does not really reflect the content of the manuscript. For example, it would be useful for readers to understand from the title that the manuscript is exclusively about the Mediterranean; 'the sea' suggests otherwise.
- line 10: would be good here to also mention a timescale
- line 26: The paper from 2015 of course is not 'recent'; especially not in ocean plastic research which is a rapidly growing field
- line 26: 'proclaimed' is perhaps a bit of a strong word here? As if politicians have planted a flag?
- line 30: Not sure that the cyclonic circulation propels debris shoreward; is there a reference for that? The inertia of most plastic seems far too small for there to be any centripetal force
- line 35: I would say that the main reason is lack of Ekman convergence?
- line 37: How does the 'crossroads' concept compare/contrast with FTLE analysis? It may be good to briefly mention that?
- line 69: I don't believe there are only two parameters. There are also parameters like the advection dt, the release interval, the output saving interval etc. These two listed here may be the most important (in

terms of sensitivity of the results), though

- line 81: A bit confusing whether the Adriatic Sea was not included in the simulation (i.e. no particles released) or in the analysis only
- line 82: normalised to what?
- line 93: although it's in the methods section, it would be good to here also mention what the grid size is
- line 135: Is there any way to validate these crossroads locations?
- line 192: How would these numbers change if biofouling is allowed to vary spatially? or is that beyond scope?
- line 337/338: Why is the number of storms not reported? Why not add it to the supplementary material?
- line 375: Does this statement mean that 90% of all plastic biofouls within 7 days?
- line 433: Are the authors planning to release animations of their simulations too?
- Figure 1: the blue/green color bar is missing? And on line 529, it should be 'purple lines'
- Figure 2: The g/km² unit for in situ observations is not used in this figure
- The captions of Figs 4 and 5 are swapped

Best wishes

Erik van Sebille

Reviewer #2 (Remarks to the Author):

This manuscript describes a numerical modeling study on floating (plastic) debris that has some novel elements in its design, and that aims to identify "crossroads", or regions with high fluxes of debris (defined as locations through which a high proportion of debris trajectories pass), rather than traditional accumulation regions (where debris resides for a relatively long period of time). I like the concept of "crossroadness" and its potential utility in identifying locations where debris might be efficiently intercepted, or with potential application to assessing wildlife exposure (where exposure at a given time might be relatively low, but cumulatively high). I think the authors could expand a bit more on these potential applications.

I am not qualified to critically assess the technical details of the numerical modeling. I do think the incorporation of shoreline steepness as a factor in beaching probability is interesting and useful, as is the incorporation of synoptic storm events. I do wonder about the probability of wash-off being dependent on length of time particles have been beached, specifically in storm events, which are anomalous events that likely could resuspend even long-beached debris. The current formulation may not be representative in storm event scenarios.

I do find some concerning discrepancies in the way "plastics" are being conceived in the model, including their input mechanisms and behavior as passive surface tracers, and the comparison to "real-

world" observations of floating plastic debris in multiple size classes. First, I think it is totally appropriate to model microplastics as passive tracers of surface circulation, since they are relatively small particles that are unlikely to be influenced directly by surface winds (i.e., via windage). However, the sources are being modeled using: 1) mismanaged waste as a proxy (for city sources), which is composed of large, everyday items found in municipal solid waste; 2) riverine input, which includes microplastics and macroplastics; and 3) input associated with ships, which is generally unknown in size and quantity. It's not clear that the proportions of microplastics being input to the Med model follow from the source assumptions being made (understanding that some assumptions must be made). Further, the correlations used to assess model performance from field-based measurements (from manta trawls with 0.333 mm mesh net) show low R/R2 values for the measured concentration by particle count, and are only significant for mass concentrations. However, mass concentrations are likely biased towards larger objects (not microplastics), that could behave differently than passive tracers. And, the lowest correlations are for the measurements of particles < 5 mm (microplastics). This does not give me much confidence in the applicability of the model beyond being a theoretical/hypothetical tool (which is still useful, but must be properly described as such).

I think this paper does have many original elements and could be of significance to the field, but I recommend a major revision addressing my major concerns above, and more detailed concerns listed below.

Lines 48-58: Can you please explain/justify the proportions used for each source (50%/30%/20%), beyond citing a prior study that did this? How dependent are the results on this assumed source distribution?

86-91: Yes, this is possible, but it is also possible that the observations have this pattern because of larger (heavier) objects occurring close to land. I think the observational data need to be included, to show the distribution of the various size classes and/or "categories" being used for model comparison.

138-145: I don't understand the value of comparing accumulations in the GPGP to crossroads in Med. Isn't the main premise of your study that you are presenting a fundamentally different measure (i.e., "crossroads", or flux, instead of accumulation)? If one were to look at crossroads in GPGP that might be interesting, but otherwise I think you are comparing apples to oranges. I suggest omitting this entirely since it is confusing and potentially misleading.

149-150: This seems to be a statement of the obvious - that crossroads will occur closest to major sources (since some measurable proportion of particles originate and must transit through locations near sources) and where ocean currents carry them (presumably, convergence zones?). Of course coastal currents must be important if 80% of debris is coming from land. I think the more important result is that 67% of particles never left the coastal buffer region. Although when you say "retained nearby", do you mean within 11 km of the point source, or within 11 km of the coast (allowing for longshore transport)? This is an important distinction and result.

161-162: Why convert this to a mass that seems essentially arbitrary?

167-170: I strongly suspect it is entirely coincidental that absolute beaching rates from observations and the model agree exactly, given the model assumptions about sources and a near-certain mismatch between size of debris collected on Corfu beaches and that represented in the model (theoretically microplastics, but correlations indicating a stronger relationship to macroplastics that may not be transported as passive tracers of surface currents), in addition to factors you discuss in the next sentences. Unless you have some way to normalize this comparison, I suggest omitting it.

173-174: I think "qualitative agreement with such measurements across the basin" is overstating the result in S.9, which is that the seasonal distribution of beaching generally agrees between the Montalto litter deposition study and model (although this might also be coincidental rather than dynamical - e.g., if beach cleanups occur more regularly in summer).

188-189: So is "sinking rate" mainly a function of time since entry (or "age")? Is it correct that every particle reaching $t = \text{"biofouling time"}$ (i.e., that has not beached before then) will sink? This is an interesting exercise, but I'm not sure what insights are gained here, given the necessarily gross source input and biofouling assumptions that must be made. Further, I do not think you can even qualitatively compare these results to seafloor surveys, since most of what is identified in these surveys is comprised of large, negatively buoyant objects - not at all comparable to the initially floating debris that you are modeling (plus all the assumptions already mentioned).

368-370: As you mention, the main objective of the paper is to identify potential "crossroads", and I wonder if, given the massive uncertainties and challenging interpretation of the results, and the fact that the TrackMPD model was run without explicitly including biofouling, you might do better to leave out the biofouling experiments here and report on those in a subsequent paper.

206-207: Without more context and information, the sentence about "similar to previous estimates" is meaningless. Was this a modeling study with similar parameters, assumptions, etc.? How were the results similar or different and why do you think this is?

S1: You state that you are modeling microplastics input, however the mismanaged waste source is better applied to macroplastic, and the high correlations apply only to larger plastics (categories 2,3 and probably 7,8). This mismatch is even more pronounced by the fact that you are using measurements of plastic mass and not particle count.

Fig 1 caption: "blue dashed" should be "magenta dotted" lines (I think).

Fig 3: Cannot see white/black circles, only numbers. I think this is an interesting figure, but it seems to be strongly related to presumed large coastal inputs, perhaps with the exception of #6 (unless there was input from Mallorca). It would be helpful to include a figure in the Supplemental Material depicting source distribution, especially for ship inputs. Please label locations of all cities mentioned in text.

Figure 4, 5 captions are swapped.

Figure 4 (beaching/sinking map) - please label geographic locations discussed in text.

Fig S1: Cannot see symbol shapes inside rectangles - too small. This might be too much to include in one figure - if there useful information about the importance of the basin of origin, I suggest making a second figure with bars shaded according to this parameter.

Reviewer #1 (Remarks to the Author):

Review of "The streaming of plastic at sea" by Baudena et al

In this article, the authors combine an extensive observational dataset, state-of-the-art simulations and a neat novel Lagrangian analysis tool to investigate the dynamics of floating plastic in the Mediterranean. This is a very relevant, well-written and original manuscript, that further pushes the understanding of an important question: the plastic floating in the Mediterranean Sea. I therefore endorse this manuscript, but encourage the authors to make a few changes for better readability

Major comments:

1) I do not think the comparison to the plastic concentration in the North Pacific makes much sense. The authors are comparing a stock to a flux, which is a bit like apples and oranges. If they really want to make this comparison then I suggest they also calculate crossroads in the Pacific; otherwise I would strongly advice to leave out of the manuscript here. There is plenty of other good material anyways

Firstly, we would like to thank Reviewer 1 for his constructive and detailed criticisms, which helped us to improve the quality of our work. In particular, we appreciated the accuracy of his remarks and the ideas suggested. We acknowledge that this comparison was unclear and potentially misleading, as it concerns a stock and a flux. For this reason, we removed it from the abstract. The reason for which we put it in the first place was because one could question whether is 20% really a significant amount of plastic debris transiting in 1% of the Mediterranean surface. This comparison would allow us to address this question, because 20% of the floating plastic in the Pacific Ocean is contained in the GPGP, which is the most notorious plastic polluted region in the world. In other words, in 1% of the Mediterranean Sea surface, 20% of plastic debris flows, while in 1% of the Pacific Ocean surface, 20% of the Pacific floating plastic debris is found. We believe that, despite not being quantitative, this is an intuitive result which could promote the understanding and the application of the crossroadness methodology. For these reasons, we modified the paragraph at L 143-156:

“The crossroadness allowed us to predict the locations through which high fluxes of plastic debris are expected to pass. The first 20 crossroads intercepted overall '13% of the virtual particles (Fig. 5) while only covering '0.3% of the Mediterranean Sea surface. The first 60 crossroads intercepted '21% of Mediterranean plastics in less than 1% of its surface. To gain a **qualitative** insight of these percentages, we note that **'1% of the Pacific Ocean surface (the Great Pacific Garbage Patch region (12,26)) contains around 18% of the estimated 117,000 tons of floating plastic in the Pacific (3). At a global level, the plastics in the Great Pacific Garbage Patch represents '8% of the estimated total floating plastic debris in around 0.4% of the surface of the world oceans. Even if the crossroadness represents a plastic flux (and not a stock as in the Great Pacific Garbage Patch), and we are comparing relative rather than absolute quantities, these values gives an idea of the magnitude of particles flowing in the Mediterranean crossroads.** Thus, even if persistent plastic accumulation zones are not present in the Mediterranean, we have shown here that a different type of structure seems to exist, crossroad regions through which large amounts of plastic debris transit”

We hope that it is now clearer and accurate.

2) The relative weighing of the different sources seems a bit arbitrary and ad hoc. It doesn't matter too much for the crossroads analysis itself, I presume.

We thank the Reviewer for raising this question. Indeed, we agree with him that we did not provide any sensitivity test of our results with respect to the proportion of particles released from cities, rivers, and vessels (pC, pR, and pV). For this reason, we carried novel analyses, resulting in three new figures added in the Supplementary Text. We used four different cases (Supplementary Fig. 5, upper table), motivating their choice at L 35-37 of the Supplementary Text

“Case 1 (40-40-20%) is usually assumed in the literature (1), Case 2 (62-32-6%) was recently estimated by (2) for the Mediterranean Sea, while Cases 3 and 4 represent intermediate proportions.”

and mentioning this analysis and its results at L 71-72, L 228-229, and L 237-238 of the main text. The first new figure (Supplementary Fig. 5) reports the correlation between observed and simulated plastic concentrations when changing pC, pR, and pV. It shows that (i) the correlation is significant for all the 4 cases studied. This proves that the model is robust with respect to changes in pC, pR, and pV; (ii) nevertheless, the best correlations are those found with pC, pR, and pV set to 50-30-20 % (Supplementary Fig. 4, upper panel), which are the values used in the main text. This corroborates the 50-30-20 % choice. Concordantly with these results, neither do the crossroadness analyses change significantly when changing these proportions (Supplementary Fig. 13). The crossroadness is lower for the Case 2 (62-32-6%), coherently with a lower input from vessels, but, importantly, the pattern remains the same. This is a notable result because we aim to assess robustness in the pattern: the relative values could change also due to a different amount of plastic entering the Mediterranean (see below), but not the main picture. Notably, the crossroad disposition does not change significantly, but only the crossroad ranking does. This analysis further highlights the robustness of the methodology introduced in the present manuscript, suggesting a potential sound applicability to other case studies.

But the weighing would matter for the analysis of total amount of beached and biofouled plastic.

Finally, we analyzed how the beaching and the surface sinking rates change. The results (Supplementary Text S.8.2 and Supplementary Fig. 19) indicate that the beaching rate is not significantly affected by the choice of pC, pR, and pV. The surface sinking rate is lower for the Case 2 (62-32-6%), as less particles were released from vessels, but the pattern remains identical. We believe that this result validates the soundness of these metrics, because, being the quantity of plastic entering the Mediterranean Sea still uncertain (see also answers below), it is the pattern that should not change.

It is also somewhat confusing that the crossroads analysis is unites (i.e. only based on particle counts), whereas the later analysis is in mass units. Why is that? Why not choose either particles or mass throughout the manuscript?

We agree with the Reviewer that the lack of homogeneity in the units is confusing and potentially misleading. The conversion from crossroadness expressed in percentage to crossroadness expressed in mass

unit is relatively easy. For this reason, we added a sentence in the main text explaining how to carry it, and providing an example (L 158-161):

“These percentages can be converted into mass fluxes, by multiplying them by the amount of plastic entering the Mediterranean Sea each year (e.g. 100,000 tons (15)) and the number of years particles were released (4). Thus, a crossroadness value of 1% would mean 4,000 tons transiting during the simulation period.”

In this way, we homogenized the units across the text. However, we preferred to keep expressing the crossroadness as a percentage. This is because (i) its pattern does not depend from the amount of plastic entering the Mediterranean Sea each year; (ii) the conversion to mass is easy (see L 158-161); (iii) it is a novel metric, and we believe that in this way its understanding is more straightforward. Finally, the surface sinking and beaching rates were kept in mass units in order to facilitate the comparison with other works (such as Kaandorp et al. 2020 or Liubartseva et al., 2018) reported at L 224-226 or in Supplementary Text S.3.

And why do the authors pose that 100,000 tons of plastic enters the ocean, when other papers give other estimates?

We acknowledge that other estimates are available, such as the one of Kaandorp et al., (2020) which estimated a maximum load of 3,200 tons entering the Mediterranean each year. The estimate we used was adopted in a previous study by Liubartseva et al. (2018), which in turn used the estimates of Jambeck et al. (2015, *Science*). The latter work estimated between 5 to 13 million metric tons entering the ocean in 2010, with a trend expected to grow exponentially by 2025. Recently, Borrelle et al. (2020, *Science*) estimated around 19 to 23 million metric tons entered aquatic systems in 2016, with 53 million tons predicted in 2030. This increasing trend has been confirmed also by Lau et al. (2020, *Science*). Therefore, the estimate of 100,000 tons/y seemed to us a good compromise between a lower (Kaandorp) and upper limit (Borrelle and Lau). All in all, we agree with the Reviewer that further discussion about this topic was needed. At the same time, this choice would not affect our results, as the pattern of our results would not change, which is the most important from our point of view. For this reason, we added the following sentences in the main text (L 174-178):

“We considered the number of particles released N as representative of 100,000 metric tons of plastic released per year. This value has been adopted in a previous study focused in the Mediterranean (15), and represents a compromise between recent estimates, both lower (27) and larger (28,29). This choice does not affect the pattern obtained (MM 3.2.5 and 3.2.6).”

L 422-425:

“Note that the surface sinking rate pattern does not depend on the amount of plastic entering the Mediterranean, but only its intensity does. For instance, assuming a 30-fold decrease (27) or 2-fold increase (28) would change the surface sinking rate proportionally, but not its pattern.”

and L 439-440:

“As for the surface sinking rate, the net beaching rate pattern does not depend on the amount of plastic entering the Mediterranean.”

Why not estimate the total flux by comparing their own model and observations?

We agree with the Reviewer that this could be a very interesting research axis. However, we think that the 122 stations we dispose of, despite constituting the largest dataset ever collected in the Mediterranean to date, still represents a limited number of observations for this type of analysis, which would need more observations. Note also that no measures were taken in the Adriatic Sea (in order not to interfere with an ongoing campaign). Nevertheless, we think that this could be an interesting analysis which we would like to conduct in the future, perhaps in a further paper. For this reason, we added a sentence in the main text (L 255-256):

“In addition, further data are crucial to improving the calibration of the model (for instance, with a bayesian approach (27)) and the description of beaching and washing-off dynamics”

3) Do the authors have any idea why their model agrees (to within error bars) with the observations in all basins except the Eastern basin? What might be going wrong in the Eastern Basin?

We agree with the Reviewer that the observed and simulated concentrations in the eastern Mediterranean are different. We think that this may be due to the fact that several land sources were present in the model in the eastern Mediterranean, in particular rivers. Another reason could be the fact that the sampling stations in this basin were distant from the shore, hampering the comparison. For this reason, we have added a sentence about this topic in the main text (L 95-97)

“The difference between the observed and simulated concentrations in the eastern Mediterranean could be due to the high number of riverine sources used in the model (21), combined with the large distance of the sampling stations from shore.”

However, while we acknowledge that this difference is significant, we also note that the eastern Mediterranean concentrations were the lowest both in observations and simulations. Furthermore, the model was not calibrated using observations, but only using realistic, non-fitted parameters, despite the multiple uncertainties still affecting the modeling of plastic debris. For these reasons, we believe that the capacity of the model in reproducing observations is a key result of our analyses.

Minor comments:

- line 1: The title does not really reflect the content of the manuscript. For example, it would be useful for readers to understand from the title that the manuscript is exclusively about the Mediterranean; 'the sea' suggests otherwise.

We acknowledge that the title was not enough specific. For this reason, we changed the title of our paper to “The streaming of plastic in the Mediterranean Sea”

- line 10: would be good here to also mention a timescale

We agree with the Reviewer that a timescale was missing. For this reason, we modified the abstract (L 9-11):

“Around 20% of Mediterranean plastic debris **released every year** passed through about 1% of the basin surface.”

- line 26: The paper from 2015 of course is not 'recent'; especially not in ocean plastic research which is a rapidly growing field

- line 26: 'proclaimed' is perhaps a bit of a strong word here? As if politicians have planted a flag?

We acknowledge that the sentence was overstating the concept, and that the mentioned paper is not recent. For these reasons, we reformulated the sentence in this way (L 28-29):

“The plastic pollution in the Mediterranean Sea is comparable to that in the major plastic accumulation zones (13), making it highly contaminated”

we hope that in this way it is clearer and more precise

- line 30: Not sure that the cyclonic circulation propels debris shoreward; is there a reference for that? The inertia of most plastic seems far too small for there to be any centripetal force

- line 35: I would say that the main reason is lack of Ekman convergence?

We address these two comments jointly. We thank the Reviewer for the remark. Indeed, we realized that the explanation based on the cyclonic circulation could not be verified based on the literature. For this reason, we removed it (L 32-33). We did not add the lack of Ekman convergence as we could not find references for that.

- line 37: How does the 'crossroads' concept compare/contrast with FTLE analysis? It may be good to briefly mention that?

We thank the Reviewer for the remark. Indeed, Lyapunov Exponents are usually used to identify barriers to transport or fronts, which can act as attracting zones (also called manifolds). The comparison of Lyapunov Exponents with the crossroadness has been the subject of our previous work (Baudena et al., 2018 doi:10.1016/j.jmarsys.2018.12.005). We report here the main results of that analysis (Fig. 4A—B):

The left panel shows the sum of the forward and backward in time Lyapunov exponents, identifying the stable and unstable manifolds of the flow. The right panel shows the crossroadness, with the black dots identifying the main crossroads (the first five in order of importance are indicated by a white number). The velocity fields is represented by the vorticity (Fig. 3)

and is characterized by 4 vortices rotating clockwise (positive vorticity, red patches) and a central vortex rotating anti clockwise (in blue). Boundary conditions are periodic. As one can see, the Lyapunov exponents and the crossroadness pattern are clearly different. This is also valid for the location of crossroads, (citing from Baudena et al., 2018):

“Although in general FTLE ridges have large crossroadness values, only the first two crossroads (white numbers 1 and 2) fall over a manifold. This fact can be understood noting that multiple crossroads over the manifolds would be redundant, because they would sample trajectories intercepted by other crossroads.”

Fig 4B shows the crossroadness calculated by advecting the whole domain. However, the crossroadness is intimately linked to the choice of the “target particles” to be monitored (in the present manuscript, represented by the virtual plastic debris; in Baudena et al., the *initialisation grid*). When advecting only a part of the domain (e.g., an ellipsoid), the difference with the Lyapunov Exponents is even more striking and not trivial (Fig 4C):

Further details can be found in Baudena et al., 2018. In conclusion, we think that mentioning the Lyapunov Exponents is out of the scope of the present manuscript because (i) Lyapunov Exponents and crossroadness are independent and different metrics, and (ii) their comparison has already been the subject of a prior publication.

- line 69: *I don't believe there are only two parameters. There are also parameters like the advection dt , the release interval, the output saving interval etc. These two listed here may be the most important (in terms of sensitivity of the results), though*

We completely agree with the Reviewer and we acknowledge that there are by far more parameters. In addition, after having carried out the sensitivity test with respect to the proportion of particles released by cities, rivers, and vessels, we realized that these are also important parameters. For these reasons we have changed the sentence in this way (L 71-72):

“The simulations therefore **mainly** depended on the two parameters K_h and TW , **and on the proportion of particles released from cities, rivers, and vessels (p_C , p_R , and p_V)”**

We also modified the relative paragraph in Materials and Methods 3.2.7 (L 458-459):

“**The sensitivity of the results with respect to the proportion of particles released from cities, rivers, and vessels (p_C , p_R , and p_V) and to the biofouling process were simulated offline.**”

- line 81: *A bit confusing whether the Adriatic Sea was not included in the simulation (i.e. no particles released) or in the analysis only*

We thank the Reviewer for this observation and we agree with him that in the old version the sentence could be misleading. For this reason, we edited it and, in addition, we moved it before. We think that now the paragraph is more understandable (L 80-86):

“To validate the model, the observed and simulated particle concentrations in the different Mediterranean sub-basins (the western Mediterranean, the Tyrrhenian and Aegean Seas, the central and the eastern Mediterranean; Fig. 1) were compared at the Tara sampling-station locations. The Adriatic Sea was not included, as no *in situ* observations were carried out in this basin. Simulated concentrations were calculated ...”

- line 82: *normalised to what?*

We agree that the sentence could be confusing. For this reason, we added a sentence at L 86-88:

“Both observed and simulated concentrations were normalized to permit comparison, **so that the sum of all the observed (or simulated) concentrations per basin was equal to 100.**”

- line 93: *although it's in the methods section, it would be good to here also mention what the grid size*

is We agree, we add the grid size at L 102:

“ For this purpose, a circular neighborhood of radius $\sigma=11$ km was defined around each point of the domain, **disposed on a regular grid of '15.7 km cell size (MM 3.3)**”

- line 135: *Is there any way to validate these crossroads locations?*

We think that two methods could be useful for that purpose. The first has probably the most interesting applications and, at the same time, more challenges. It would be the use of a fixed station (e.g. a mooring) with a manta net, oriented perpendicularly to the current direction, constantly intercepting plastic debris. The challenge of this method would be both structural, both linked to the necessity of a regular removal of the plastic debris from the manta net. Also, the radius of the circular neighborhood used in our simulation (11 km) is by far larger than the eventual size of the manta net (in the order of meters). Our analyses have shown a robustness of the crossroad location with respect to the circular neighborhood radius, but further studies are necessary to test it at smaller scales. The second method is probably the easiest to concretise. It would consist of the use of a fleet of drifters (e.g. SVP drifters, which float relatively close to the surface). By analyzing their trajectories, one could calculate the location of the 'drifter crossroads'. However, we think that (i) a large number of drifters would be necessary; (ii) the release locations should be carefully planned, in order to represent the distribution of the different plastic sources. Nevertheless, since the use of drifters for oceanographic purposes has increased exponentially in the last years, we believe that this is the most easy solution, which can be the subject of future studies.

- line 192: *How would these numbers change if biofouling is allowed to vary spatially? or is that beyond scope?*

We thank the Reviewer for the interesting suggestion. Indeed we think that this could be a promising exploration axis for the future, and which could be subject of a future paper. For these reasons, we added a sentence in the main text about this concept (L 261-262):

“The description of surface sinking rates could benefit from considering spatial variation in biofouling rates (34,43).”

- line 337/338: Why is the number of storms not reported? Why not add it to the supplementary material?

We agree with the Reviewer that this is an interesting result worthy of publication. For this reason, we calculated the number of the storm (washing-off) events on all the Mediterranean shores during the simulation period (2013—2017) as a function of the month considered. Subsequently, we normalized it by dividing the number of storms in each month by the total number. We reported the results in Supplementary Fig. 3, and we mentioned it across the main text (L 375-378):

“The number of storms per month obtained in this way (Supplementary Fig. 3) was in line with the beaching and resuspension events observed on the Israeli coast (25). Storm frequency has a seasonal cycle: an increased number of events occurs during winter and early spring, fewer during summer”

- line 375: Does this statement mean that 90% of all plastic biofouls within 7 days?

No, it does not, and we thank the Reviewer for making us notice that the sentence was written in a misleading way. First, we edited the sentence (L 426-429, reported below). Secondly, we added a plot representing the fraction of biofouled mass as a function of the time spent in water (Supplementary Fig. 2):

“We set $r=1/3$, so that the shape of the probability function curve was consistent for the four T_{BF} values (Supplementary Fig. 2). With this r value, the fraction of biofouled mass increased from 0.1 to 0.9 in a period of time of ~ 7 days centered on T_{BF} . ”

We added reference to this new figure also in the main text (L 203-205 and L 207-209) and in MM 3.2.5 (L 404-406):

“ As the time spent in the water approached the biofouling time, the fraction of the mass which sank increased exponentially (Supplementary Fig. 2)”

“For these reasons, we used four biofouling times (from 50 to 200 days, Supplementary Fig. 2), and averaged the results”

“Thus, based on Eq. (2), we could estimate the amount of plastic biofouled at each time step of each trajectory (Supplementary Fig. 2)”

- line 433: Are the authors planning to release animations of their simulations too?

We acknowledge that this could be an added value of the paper. However, we prefer not to include an animation of our simulation for two main reasons: (i) we believe that the representation of the key concept of our paper, the plastic crossroad, is already well achieved by Figs. 3B and 3C; (ii) the manuscript in its present version counts 24 Figures (5 in the main text, 19 in the Supplementary Text), so it is already dense in contents. However, we will consider producing animations for future works (or seminars).

- Figure 1: the blue/green color bar is missing?

We decided to represent all the cities (and all the rivers) with the same color (blue and green, respectively), and not proportionally to the amount of plastic released because this would add a level of information which we believe not necessary (please note that there are already two colorbars in Fig. 1). In addition, the location of all the plastic sources as well as the proportion of particles they release will be publicly available.

And on line 529, it should be 'purple lines'

We corrected, thanks (L 619)

- Figure 2: The g/km² unit for in situ observations is not used in this figure

We agree, we removed it from Figure 2 legend.

- The captions of Figs 4 and 5 are swapped

It is true, thanks! We swapped them.

Best wishes

Erik van Sebille

Reviewer 2

This manuscript describes a numerical modeling study on floating (plastic) debris that has some novel elements in its design, and that aims to identify "crossroads", or regions with high fluxes of debris (defined as locations through which a high proportion of debris trajectories pass), rather than traditional accumulation regions (where debris resides for a relatively long period of time). I like the concept of "crossroadness" and its potential utility in identifying locations where debris might be efficiently intercepted, or with potential application to assessing wildlife exposure (where exposure at a given time might be relatively low, but cumulatively high). I think the authors could expand a bit more on these potential applications.

First of all, we would like to thank the Reviewer 2 for its constructive criticisms and advice, and we are glad he/she appreciated the concepts we introduced in the present manuscript. We believe that the soundness of our results could foster the application of the crossroadness methodology to other case studies, other pollutants, or to assess wildlife exposure as highlighted by the Reviewer. In this context, we agree with him that the potential applications of the crossroadness concept should be stressed more. For this reason, we edited the perspective paragraph in the main text as follows (L 264-269):

"Finally, the ecological implications of the presence of crossroads should be investigated. Although they are not necessarily zones of high plastic concentration, they are regions in which wildlife exposure to plastic at a given time might be relatively low, but cumulatively high. For example, in fish nursery areas, the presence of toxin-contaminated plastics can impact the survival of fish larvae during this very vulnerable stage of their life cycle, with significant socio-economic consequences (44)"

and also at L 245-248:

"At regional scales, this method could help design a network of fixed stations to monitor and eventually remove plastic items from the sea surface (e.g. <https://theoceancleanup.com/>), especially in basins where plastic debris does not accumulate (37—40)"

I am not qualified to critically assess the technical details of the numerical modeling. I do think the incorporation of shoreline steepness as a factor in beaching probability is interesting and useful, as is the incorporation of synoptic storm events. I do wonder about the probability of wash-off being dependent on length of time particles have been beached, specifically in storm events, which are anomalous events that likely could resuspend even long-beached debris. The current formulation may not be representative in storm event scenarios.

We thank the Reviewer for the interesting comment. We agree with him that the probability of washing-off is still uncertain, as almost no studies investigated this dynamics. Furthermore, the majority of studies modelling plastic debris dispersion did not even include beaching, including recent ones in the

Mediterranean Sea (e.g. Mansui et al., 2020; doi:10.1016/j.pocean.2020.102268). To our knowledge, washing-off in the Mediterranean Sea has only been included in the model of Liubartseva et al. (2018; doi:10.1016/j.marpolbul.2018.02.019). There, washing-off events could occur at any time, based on a Monte-Carlo process. In our work, we tried to improve the washing-off characterisation while keeping the assumptions reasonable and the numerical calculation efficient. We used one of the few studies available (Bowmann et al., 1998, *Journal of Coastal Research*), which highlighted that storms are the main responsible for the washing-off of beached debris. They also observed a strong turnover of marine litter on Mediterranean beaches. However, they did not establish whether the removal of debris from a beach was due to a storm which resuspended it at sea again, or to other reasons (e.g. the wind which moved it away or buried it). We believe that a consistent fraction of the debris was reabsorbed at sea due to the storms. At the same time, we think that other factors such as wind-induced movement and burial can reduce the probability of washing-off as the time spent on the beach increases. Wind-induced dispersion of plastic toward the top of the beach has already been observed in Atlantic beaches (Lefebvre et al., 2021; doi:10.1016/j.scitotenv.2021.149144) and suggested in many other studies (e.g. Browne et al., 2010; doi:10.1021/es903784e), while microplastic debris has been found buried in Mediterranean beaches (e.g. Ceccarini et al., 2018; doi:10.1021/acs.est.8b01487). Clean-up activities may also remove plastic from beaches. In addition, Liubartseva et al., (2018) used a washing-off probability decreasing with the time spent on the beach as well.

Finally, and most importantly, the crossroadness does not change significantly when the mean time spent by a particle on the beach TW increased by 50 to 200 days (please note that as TW increases, the closest we are to the hypothesis suggested by the reviewer, i.e. that the probability of washing-off does not change with time):

The 4 plots show the crossroadness calculated while using 4 different values of TW, from 50 to 200 days. The crossroadness pattern and the crossroads disposition remain almost identical.

In conclusion, (i) while the current literature cannot exclude the hypothesis suggested by the Reviewer, it seems reasonable that the probability of resuspension decreases as time spent on the beach increases; (ii) in any case, the results would not be significantly affected by this choice.

I do find some concerning discrepancies in the way "plastics" are being conceived in the model, including their input mechanisms and behavior as passive surface tracers, and the comparison to "real-world" observations of floating plastic debris in multiple size classes. First, I think it is totally appropriate to model microplastics as passive tracers of surface circulation, since they are relatively small particles that are unlikely to be influenced directly by surface winds (i.e., via windage). However, the sources are being modeled using: 1) mismanaged waste as a proxy (for city sources), which is composed of large, everyday items found in municipal solid waste; 2) riverine input, which includes microplastics and macroplastics; and 3) input associated with ships, which is generally unknown in size and quantity. It's not clear that the proportions of microplastics being input to the Med model follow from the source assumptions being made (understanding that some assumptions must be made).

We thank the Reviewer for the interesting comment and we acknowledge that there was a discrepancy between the plastic simulated in the model and the plastic observed in situ. Indeed, we recognize that, in the way it is built, our model also simulates plastic debris larger than 5 mm. The plastic sources considered (city, rivers, and vessels) release also larger items; this is perfectly consistent with the fact that the correlations between simulated and observed plastic are larger when considering all size classes. The model can simulate the transport of debris larger than 5 mm, even if they are close to the surface, because the velocity field used takes into account the Stokes drift. The Stokes drift is the movement due to waves, and it is a fundamental component of the velocity field to be taken into account when modelling plastic debris, as it includes indirectly the windage effect (L 64-66). The recent study of Onink et al., (2019; doi:10.1029/2018JC014547) highlighted that the use of Stokes drift provides a better description of plastic dispersion rather than the windage. This is because there is a time lag between the presence of windage and the consequent wave formation. For these reasons, we think that our model can simulate efficiently a larger size spectra of particles than previously mentioned. However, we agree with the Reviewer that the windage can affect plastic transport importantly when very light plastic items and air-filled objects are considered. Therefore, we changed the main text at L 51-54 as follows:

“ The simulated particles are considered representative of floating plastics debris, with the exception of extremely light foamed plastics (such as polystyrene foam) or air filled objects (such as bottles or balls) whose dispersion is mainly driven by windage (19). These were less than 1% of the debris collected during the Tara Expedition.”

Further, the correlations used to assess model performance from field-based measurements (from manta trawls with 0.333 mm mesh net) show low R/R2 values for the measured concentration by particle count, and are only significant for mass concentrations.

We acknowledge that the correlations for the measured concentration by particle count are lower than those by particle mass. However, this is not surprising, due to two main reasons: first, current literature reports the amount of plastic entering in the Mediterranean as weight, and not by counts. Coherently, in our model, the amount of particles released from each city or river was set proportional to the plastic mass released by that city or river as reported in the literature. For the cities, we used the estimates of Jambeck et al. (2015; doi:10.1126/science.1260352) which reported the plastic mass released by each country in the Mediterranean. For the rivers, we used the dataset of Lebreton et al. (2017; doi:10.1038/ncomms15611) which calculated how many kg of plastic a given river mouth releases each year. Secondly, even if we knew the amount of particles released by each source, still the use of particle counts would be biased by the fragmentation process. This dynamic is actually poorly documented and can not be included in plastic modelling studies to date. The bias introduced by such a process can be great: for instance, a plastic bottle can break in more than 200 (20,000) debris of 1 cm (mm) size. However, the fragmentation conserves the mass of the plastic. For these reasons, we are not surprised to obtain the best correlations with the measured concentration by mass and not by particle count. Finally, we note that the correlations with the measured concentration by particle count, even if not significant, still display large R values (> 0.8 , upper panel of Supplementary Fig. 4)

However, mass concentrations are likely biased towards larger objects (not microplastics), that could behave differently than passive tracers. And, the lowest correlations are for the measurements of particles < 5 mm (microplastics).

See answer above

This does not give me much confidence in the applicability of the model beyond being a theoretical/hypothetical tool (which is still useful, but must be properly described as such).

For the reasons listed above, we believe that the model presented in this manuscript goes beyond a purely theoretical approach. Please also note that this model was not directly calibrated with observations, but that we used realistic, non-fitted parameters, and it has been the first to be quantitatively validated in the Mediterranean Sea. In addition the results are robust to a large variety of parameters used. Therefore, we think that this model can also have concrete and practical applications. We have stressed this more in depth in the main text (L 241-248):

“The adaptability and accuracy of the method here could lead to its application to other regions and time windows, or into the study of other pollutants or passive tracers (36). At a global level, comparison of the plastic crossroads with the major world plastic-accumulation zones could add to knowledge on plastic transport, spreading, and accumulation. At regional scales, this method could help design a network of fixed stations to monitor and eventually remove plastic items from the sea surface (e.g. <https://theoceancleanup.com/>), especially in basins where plastic debris does not accumulate.”

Of course, further empirical data will allow us to improve the description of dynamics such as beaching and washing-off, and potentially to include others (such as the fragmentation). These aspects are mentioned at L 249-252 and L 255-261.

I think this paper does have many original elements and could be of significance to the field, but I recommend a major revision addressing my major concerns above, and more detailed concerns listed below.

Lines 48-58: Can you please explain/justify the proportions used for each source (50%/30%/20%), beyond citing a prior study that did this? How dependent are the results on this assumed source distribution?

We thank the Reviewer for the interesting consideration. Indeed, in the previous version, we have not tested the dependency of the results with respect to the proportions of the particles released from cities, rivers, and vessels (pC, pR, and pV). Following the hint of the Reviewer, we decided to carry out a comprehensive sensitivity test which included the comparison between observed and simulated plastic concentrations, the crossroadness, and the beaching and surface sinking rates. We used 4 different proportions, which are reported in the upper table of Supplementary Fig. 5, and which are justified based on the literature (L 35-37 of the Supplementary Text)

“Case 1 (40-40-20%) is usually assumed in the literature (1), Case 2 (62-32-6%) was recently estimated by (2) for the Mediterranean Sea, while Cases 3 and 4 represent intermediate proportions.”.

The comparison between observed and simulated concentrations was significant when changing pC, pR, and pV, which proved the robustness of the model with respect to these parameters (Supplementary Fig. 4 and Supplementary Text S.1.2). In addition, the best correlations were obtained with the proportion 50-30-20% (upper panel of Supplementary Fig. 4) which was used in the main text, further corroborating the use of such values. Neither did the crossroadness nor the crossroads were affected by the variation in pC, pR, and pV (Supplementary Fig. 13 and Supplementary Text S.6.4): the crossroadness slightly diminished in the Case 2 only (62-32-6%) due to a lower particle input from vessels, but the pattern remained the same. The location of the plastic crossroads was almost identical, and only the ranking changed. This confirmed the soundness of the crossroadness methodology with respect to these parameters. Finally, the net beaching rate did not change significantly when changing pC, pR, and pV (Supplementary Fig. 19 and Supplementary Text S.8.2). The surface sinking rate diminished in the Case 2 only (62-32-6%), due to a lower contribution of vessels, but its pattern was not affected (Supplementary Fig. 19 and Supplementary Text S.8.2). We think that these novel analyses contributed to address the question mentioned by the Reviewer. These results are now mentioned in the main text as well (L 227-229):

“The results presented here did not change significantly across the 16 scenarios simulated (Supplementary Text S.1.1, S.4, S.6.1, S.8.1), neither when changing the proportion of particles released from cities, rivers, and vessels (pC, pR, and pV; Supplementary Text S.1.2, S.6.4, S.8.2).”

86-91: Yes, this is possible, but it is also possible that the observations have this pattern because of larger (heavier) objects occurring close to land. I think the observational data need to be included, to show the distribution of the various size classes and/or "categories" being used for model comparison.

We thank the Reviewer for the interesting consideration. We agree that, in principle, larger items close to the coast could be responsible for a larger concentration observed. To test this hypothesis, we performed a novel analysis following the Reviewer hint. In situ stations were regrouped into four shore-distance classes. For each class, we calculated the size distribution by summing the size distribution of all the stations in that

class. Finally, we normalised the abundance, so that the sum of the abundances of a given class gives 1. The results are reported in Supplementary Fig. 6C. The size distributions for the 4 different categories are very similar. Only the category 40–60 km shows some difference, likely due to the low number of stations falling in that distance range (7 out of 122). The fact that the size distributions are very similar allows us to exclude the hypothesis suggested by the Reviewer: for instance, even if we are close to the coast and that we find more plastic debris, the proportion of small and large items would be the same as when we are far offshore. These results are now described in Section S.2 of the Supplementary Text (L 57-66):

“However, a larger observed concentration close to the coast could be explained by a greater number of large debris (thus heavier) occurring close to land sources. To verify this hypothesis, we calculated the size distribution for each class and we normalised it, so that the sum of the abundance of a given class was equal to 1 (Supplementary Fig. 6C). The size distribution did not change considerably when changing the shore distance. A slight difference is present in class 40–60 km, likely due to the relatively low number of in situ stations (7 out of 122). The results show that the distribution of plastic concentration as a distance from the shore is not explained by a greater occurrence of larger items close to the shore. Furthermore, they indicate the coastal area as a region of debris retention, in accordance with our results (e.g. Fig. 5, Supplementary Fig. 12).”

138-145: I don't understand the value of comparing accumulations in the GPGP to crossroads in Med. Isn't the main premise of your study that you are presenting a fundamentally different measure (i.e., "crossroads", or flux, instead of accumulation)? If one were to look at crossroads in GPGP that might be interesting, but otherwise I think you are comparing apples to oranges. I suggest omitting this entirely since it is confusing and potentially misleading.

We acknowledge that this comparison was unclear and potentially misleading, as it concerns a stock and a flux. For this reason, we removed it from the abstract. The reason for which we put it in the first place was because one could question whether is 20% really a significant amount of plastic debris transiting in 1% of the Mediterranean surface. This comparison would allow us to address this question, because 20% of the floating plastic in the Pacific Ocean is contained in the GPGP, which is the most notorious plastic polluted region in the world. In other words, in 1% of the Mediterranean Sea surface, 20% of plastic debris flows, while in 1% of the Pacific Ocean surface, 20% of the Pacific floating plastic debris are found. We believe that, despite not being quantitative, this is an intuitive result which could promote the understanding and the application of the crossroadness methodology. For these reasons, we modified the paragraph at L 143-156:

“The crossroadness allowed us to predict the locations through which high fluxes of plastic debris are expected to pass. The first 20 crossroads intercepted overall 13% of the virtual particles (Fig. 5) while only covering 0.3% of the Mediterranean Sea surface. The first 60 crossroads intercepted 21% of Mediterranean plastics in less than 1% of its surface. To gain a qualitative insight of these percentages, we note that 1% of the Pacific Ocean surface (the Great Pacific Garbage Patch region (12,26)) contains around 18% of the estimated 117,000 tons of floating plastic in the Pacific (3). At a global level, the plastics in the Great Pacific Garbage Patch represents 8% of the estimated total floating plastic debris in around 0.4% of the surface of the world oceans. Even if the crossroadness represents a plastic flux (and not a stock as in the Great Pacific Garbage Patch), and we are comparing relative rather than absolute quantities, these values gives an idea of the magnitude of particles flowing in the Mediterranean crossroads. Thus, even if persistent

plastic accumulation zones are not present in the Mediterranean, we have shown here that a different type of structure seems to exist, crossroad regions through which large amounts of plastic debris transit”

We hope that in this way the concept expressed is clearer.

149-150: This seems to be a statement of the obvious - that crossroads will occur closest to major sources (since some measurable proportion of particles originate and must transit through locations near sources) and where ocean currents carry them (presumably, convergence zones?). Of course coastal currents must be important if 80% of debris is coming from land. I think the more important result is that 67% of particles never left the coastal buffer region. Although when you say "retained nearby", do you mean within 11 km of the point source, or within 11 km of the coast (allowing for longshore transport)? This is an important distinction and result.

We thank the Reviewer for this consideration. When saying ‘retained nearby’ we mean within the buffer region (11 km from the land source). What we want to say here is that most particles do not leave the coastal region where they were released. If sources were distributed in a different way, this percentage (67%) would probably not change, because this dynamic is not due to the land source disposition, but to the Mediterranean circulation that propels debris onshore. This is confirmed by the robustness of the results with respect to changes in the proportion of particles released from cities, rivers, and vessels. We kindly disagree with the Reviewer when he/she says that the importance of coastal currents for plastic debris transport is obvious. It would be obvious if the buffer around land sources was absent. The use of the buffer prevents us from finding trivial solutions. The proof is that the first-ranked crossroad is not close to the larger land source, nor the second-ranked one and so on. Also, for instance, the first-ranked crossroad intercepts particles coming from more than 400 km away. Finally, to our knowledge, current literature does not mention the Mediterranean coastal currents as the main driver of plastic dispersion.

161-162: Why convert this to a mass that seems essentially arbitrary?

We acknowledge that this question deserved further details. First, the use of a mass to convert the number of particles into a weight has already been used in previous plastic modelling studies in the Mediterranean (e.g. Liubartseva et al., 2018; Kaandorp et al., 2020) because it allowed the authors to provide concrete quantities which can be interpreted more easily, for instance, by decision makers. Therefore our decision is driven by these reasons and by the possibility of comparing our results with these works. Secondly, the amount of plastic entering the Mediterranean, even if uncertain, is not arbitrary and has been debated in multiple scientific papers. Kaandorp et al., (2020) estimated around 3,200 tons of plastic per year. Liubartseva et al. (2018) used a value of 100,000 tons of plastic per year, which was obtained from the estimates of Jambeck et al. (2015, *Science*). Jambeck et al. estimated between 5 to 13 million metric tons entering the ocean in 2010, with a trend expected to grow exponentially by 2025. Recently, Borrelle et al. (2020, *Science*) estimated around 19 to 23 million metric tons entered aquatic systems in 2016, with 53 million tons predicted in 2030. This increasing trend has been confirmed also by Lau et al. (2020, *Science*). We decided to use 100,000 tons per year as it represented a good compromise between the lower (Kaandorp) and upper (Borrelle and Lau) estimates available. Please also note that, by using the quantities estimated by Kaandorp or Borrelle, the magnitude of the surface sinking rate would change proportionally, but the pattern would not. We think that

this aspect will facilitate the use of our results, which will be publicly available. We have highlighted this question in the main text at L 174-178:

“We considered the number of particles released N as representative of 100,000 metric tons of plastic released per year. This value has been adopted in a previous study focused in the Mediterranean (15), and represents a compromise between recent estimates, both lower (27) and larger (28,29). This choice does not affect the pattern obtained (MM 3.2.5 and 3.2.6).”

and in Materials and Methods 3.2.5 (L 422-425):

“Note that the surface sinking rate pattern does not depend on the amount of plastic entering the Mediterranean, but only its intensity does. For instance, assuming a 30-fold decrease (27) or 2-fold increase (28) would change the surface sinking rate proportionally, but not its pattern.”

and MM 3.2.6 (L 439-440):

“As for the surface sinking rate, the net beaching rate pattern does not depend on the amount of plastic entering the Mediterranean.”

Finally, in order to avoid a potential confusion due to the fact that the crossroadness is expressed as percentage and not in mass units, we specified how to convert it at L 158-161:

“These percentages can be converted into mass fluxes, by multiplying them by the amount of plastic entering the Mediterranean Sea each year (e.g. 100,000 tons (15)) and the number of years particles were released (4). Thus, a crossroadness value of 1% would mean 4,000 tons transiting during the simulation period.”

This allowed us to homogenise the measure units we used across the text

167-170: I strongly suspect it is entirely coincidental that absolute beaching rates from observations and the model agree exactly, given the model assumptions about sources and a near-certain mismatch between size of debris collected on Corfu beaches and that represented in the model (theoretically microplastics, but correlations indicating a stronger relationship to macroplastics that may not be transported as passive tracers of surface currents), in addition to factors you discuss in the next sentences. Unless you have some way to normalize this comparison, I suggest omitting it.

We thank the Reviewer for the comment. We acknowledge that the sentence overstated the result we obtained. However, our previous answer addressed the “near-certain mismatch between size of debris collected on Corfu beaches and that represented in the model” question. A mismatch probably exists, but not in the magnitude suspected before, because the model can simulate also larger debris. Furthermore, the simulated beaching rate depends on the total amount of plastic entering the Mediterranean, which, despite being subject to uncertainty, has scientific bases (see answer to previous comment). For these reasons, we decided to keep the comparison with the Corfu beaching rate while declaring the associated uncertainty (L 182-187):

“The net beaching rates predicted on Corfu [(1.9±2.3) kg/km/day] were very similar to the observed values [(1.9±2.2) kg/km/day, (30)], even if the predicted values would change when changing the amount of plastic released per year. Comparing the values obtained in other regions with observations was not possible, as most of the time only occasional measurements are carried out on Mediterranean beaches.”

173-174: I think "qualitative agreement with such measurements across the basin" is overstating the result in S.9, which is that the seasonal distribution of beaching generally agrees between the Montalto litter deposition study and model (although this might also be coincidental rather than dynamical - e.g., if beach cleanups occur more regularly in summer).

We kindly disagree with the Reviewer comment. Indeed, Section S.9 reports also the qualitative agreement of the simulated and observed beaching rates for two further studies, focused on the eastern Mediterranean. In addition, note that the Montalto study by Poeta et al., 2016 was carried out following rigorous sampling methodologies. The studies area was limited to human access, and no beach cleanups were carried with the exclusion of those of the authors (citing from the text):

2. Materials and methods

2.1. Study area

A suitable study area was chosen following the criteria suggested in the GMMML. We selected a fine-grained sandy beach on the North side of the Lazio region (Central Tyrrhenian Italy) (Fig. 1A, B) near Montalto Marina (Viterbo). This area is among the least disturbed coastal zones in Central Italy. It is characterized by a relatively good conservation status, as supported by previous ecological investigations (Carboni et al., 2009; Fenu et al., 2013).

The beach is characterized by a mean slope value that ranges from 5 to 9 degrees (Bazzichetto et al., 2016) with a South-West orientation. It is 8.5 km long and is naturally delimited to the South by the Fiora river and to the North by a small creek, both characterized by varying seasonal discharge rates. The Fiora river flows mainly across cultivated lands and its riverbanks are also used as docks. The study area is protected inland by a power plant and a privately owned natural area. Therefore, human visitors are limited, throughout the year, to fishermen and a few beach tourists who visit the northern part of the beach during the summer. Moreover, the area is not subject to litter collection activities (except in the case of this survey).

For these reasons we believe that the matchup between observed and simulated beaching rate, even if not quantitative, can be defined as ‘qualitative’.

188-189: So is "sinking rate" mainly a function of time since entry (or "age")?

The sinking rate depends on the time since entry, but it is mainly a function of the time spent in water (L 203-205):

“As the time spent in the water approached the biofouling time, the fraction of the mass which sank increased exponentially (Supplementary Fig. 2)”

This means that if a particle is beached than the biofouling stops growing.

Is it correct that every particle reaching t ="biofouling time" (i.e., that has not beached before then) will sink?

We thank the Reviewer for the consideration. Indeed, the text was potentially misleading concerning this point. To help us express this concept, we added a novel plot (Supplementary Fig. 2):

Supplementary Fig. 2 shows the fraction of the biofouled mass m_B as a function of the time spent in water, for 4 different biofouling times (50, 100, 150, and 200 days). The biofouled mass m_B increases *before* and *after* the biofouling time. This permitted us to provide a more realistic description of the biofouling (which is unlikely to occur all of a sudden). We mentioned the novel figure in the main text and edited the sentence describing the biofouling (L 426-429):

“We set $r=1/3$, so that the shape of the probability function curve was consistent for the four T_{BF} values (Supplementary Fig. 2). With this r value, the fraction of biofouled mass increased from 0.1 to 0.9 in a period of time of ~ 7 days centered on T_{BF} .”

We added reference to this new figure also in the main text (L 203-205 and L 207-209) and in MM 3.2.5 (L 404-406):

“As the time spent in the water approached the biofouling time, the fraction of the mass which sank increased exponentially (Supplementary Fig. 2)”

“For these reasons, we used four biofouling times (from 50 to 200 days, Supplementary Fig. 2), and averaged the results”

“Thus, based on Eq. (2), we could estimate the amount of plastic biofouled at each time step of each trajectory (Supplementary Fig. 2)”

This is an interesting exercise, but I'm not sure what insights are gained here, given the necessarily gross source input and biofouling assumptions that must be made. Further, I do not think you can even qualitatively compare these results to seafloor surveys, since most of what is identified in these surveys is comprised of large, negatively buoyant objects - not at all comparable to the initially floating debris that you are modeling (plus all the assumptions already mentioned).

We thank the Reviewer for the interesting remark. We added a sentence about the fact that these campaigns collected also objects whose pristine density is larger than seawater (L 263-264 of the Supplementary Text):

“The debris collected were composed of plastic with a density larger than seawater, which were not modelled in the present study”

However, we kindly disagree with the Reviewer suggestion to remove these comparisons. First, we reported five independent studies realised in different part of the Mediterranean basins, in particular in the regions where the surface sinking rate is predicted to be the lowest and the highest. Secondly, it would make sense to remove them only if the matchings were absent. We agree with the Reviewer about the fact that several assumptions were made, in particular concerning the modelisation of the surface sinking rate. However, our study represents a step forward into the description of this process, and the qualitative agreement with in situ observations suggests that the direction we took is promising. In addition, please note that these results have a strong interest for the scientific community: the plot of the surface sinking rate calculated by Liubartseva et al. (2018; their Fig. 6) was used by Kane et al. (2020; doi:10.1126/science.aba5899) in their work published on *Science*. Kane et al. used the surface sinking rate as an estimate of the seafloor concentration. This shows the great interest of the scientific community for this metric, and highlights the lack of adapted estimates for the seafloor concentration.

368-370: As you mention, the main objective of the paper is to identify potential "crossroads", and I wonder if, given the massive uncertainties and challenging interpretation of the results, and the fact that the TrackMPD model was run without explicitly including biofouling, you might do better to leave out the biofouling experiments here and report on those in a subsequent paper.

See answer to the previous comment

206-207: Without more context and information, the sentence about "similar to previous estimates" is meaningless. Was this a modeling study with similar parameters, assumptions, etc.? How were the results similar or different and why do you think this is?

A detailed comparison of our and Liubartseva results is provided in Supplementary Text S.3. Therefore, we edited the sentence as follows (L 225-226):

"Local differences are explained by the diverse nature of the sources, and by the beaching and washing-off dynamics in our model (MM and Supplementary Text S.3)."

S1: You state that you are modeling microplastics input, however the mismanaged waste source is better applied to macroplastic, and the high correlations apply only to larger plastics (categories 2,3 and probably 7,8). This mismatch is even more pronounced by the fact that you are using measurements of plastic mass and not particle count.

See answer above

Fig 1 caption: "blue dashed" should be "magenta dotted" lines (I think).

Yes, thanks! We corrected it (L 619)

Fig 3: Cannot see white/black circles, only numbers

The size of the circles is proportional to the radius of the circular neighborhood used for the calculation of the crossroadness. Therefore, increasing their size is not possible. It is strange that they are not visible, we

can perfectly identify them on our pdf version. The figure was edited by a specialist so we hope that you could spot them in the present version.

I think this is an interesint figure, but it seems to be strongly related to presumed large coastal inputs, perhaps with the exception of #6 (unless there was input from Mallorca).

The sixth-ranked crossroad position was not affected by Palma de Mallorca city, which is more than 80 km away, as mentioned in the main text (L 133-136)

“In this regard, it is remarkable that the sixth-ranked crossroad, located north of Mallorca in the Balearic Archipelago (Fig. 3B,C), was situated far from all land sources (the only nearby city, Palma de Mallorca, was more than 80 km away).”

and which did not represent the main source of the particles intercepted by the crossroad (vessels, and the 3 main cities were Barcelona, Algiers, and Valencia, as mentioned at L 138-140:

“Surprisingly, 9% of these particles came from Algiers, followed by Barcelona (8.5%) and Valencia (3%), even though these cities were 380 km, 150 km, and 250 km away, respectively (Fig. 3C)”.

Concerning the observation about the location of the crossroads, we clearly mention this aspect in the main text and provide evidence of their originality (L 126-128):

“The first twenty crossroads in the rank order (Fig. 3A) were situated near the coast, generally in proximity to a land source. However, in most cases the intercepted particles were from multiple sources and had often traveled long distances (Supplementary Fig. 1)”

and L 162-166:

“The locations of the crossroads seem to be intimately connected with the anthropogenic pollution sites and with the circulation features transporting the debris. In this regard, the fact that most crossroads lay in coastal areas suggests that boundary currents play a pivotal role in determining their location. These circulation structures can collect large quantities of plastic debris released from land, funnelling and carrying them over large distances.”

See also the answer relative to the coastal currents above.

It would be helpful to include a figure in the Supplemental Material depicting source distribution, especially for ship inputs.

We think the Reviewer refers to Figure 1 of the main text, which contains the distribution of all the sources. Cities and rivers are shown as green and blue dots, respectively. The vessel input is indicated by the grey scale colouring the Mediterranean surface (grey scale at left).

Please label locations of all cities mentioned in text.

We thank the Reviewer for the suggestion. We added the name of the cities mentioned in the crossroadness analysis in Figure 3 of the main text.

Figure 4, 5 captions are swapped.

This is true, thanks! We swapped the captions.

Figure 4 (beaching/sinking map) - please label geographic locations discussed in text.

We thank the Reviewer for the suggestion. We added the name of the cities mentioned in the surface sinking and beaching rate analysis in Figure 5 of the main text.

Supplementary Fig. 1: Cannot see symbol shapes inside rectangles - too small. This might be too much to include in one figure - if there useful information about the importance of the basin of origin, I suggest making a second figure with bars shaded according to this parameter.

We thank the Reviewer for the suggestion. We have incremented the size of the symbols, their line width, and the size of the Supplementary Figure 1. We hope that now it is more visible.

REVIEWERS' COMMENTS

Reviewer #1 (Remarks to the Author):

I am happy with the changes made by the authors, and can now recommend the manuscript for publication

Reviewer #2 (Remarks to the Author):

I thank the authors for seriously and comprehensively addressing my lengthy comments on the initial submission, and especially for carrying out additional calculations (e.g., on source proportions) that strengthen the results of the work. I am happy to recommend that this manuscript now be accepted for publication.